# Evolution and expansion of Li concentration gradient during charge–discharge cycling

Byeong-Gyu Chae [1,3,4✉], Seong Yong Park [1,3,4✉], Jay Hyok Song[2], Eunha Lee [1] & Woo Sung Jeon[1]

To improve the performance of Li-ion batteries (LIBs), it is essential to understand the behaviour of Li ions during charge–discharge cycling. However, the analytical techniques for observing the Li ions are limited. Here, we present the complementary use of scanning transmission electron microscopy and atom probe tomography at identical locations to demonstrate that the evolution of the local Li composition and the corresponding structural changes at the atomic scale cause the capacity degradation of $Li(Ni_{0.80}Co_{0.15}Mn_{0.05})O_2$ (NCM), an LIB cathode. Using these two techniques, we show that a Li concentration gradient evolves during cycling, and the depth of the gradient expands proportionally with the number of cycles. We further suggest that the capacity to accommodate Li ions is determined by the degree of structural disordering. Our findings provide direct evidence of the behaviour of Li ions during cycling and thus the origin of the capacity decay in LIBs.

[1] Analytical Engineering Group, Material Research Center, Samsung Advanced Institute of Technology, Samsung Electronics Co., Ltd., Suwon, Republic of Korea. [2] Materials Development Group 1, Samsung SDI, Suwon, Republic of Korea. [3]These authors contributed equally: Byeong-Gyu Chae, Seong Yong Park. [4]These authors jointly supervised this work: Byeong-Gyu Chae, Seong Yong Park. ✉email: bg.chae@samsung.com; sydra.park@samsung.com

Rechargeable Li-ion batteries (LIBs) have attracted great interest due to their explosive increase in demand for devices ranging from small portable electronics to large energy-storage devices[1–8]. This rapid expansion necessitates further improving their performance with regard to their capacity, charging speed, lifetime and safety. Many studies have investigated the action/degradation mechanisms of LIB cathode materials to understand the underlying physics in order to develop performance improvement strategies[9–15]. However, these mechanisms were primarily proposed based on the observation of the positions and chemical states of the transition metals (TMs) and oxygen atoms, but not those of Li ions, which are directly responsible for the battery operation because of the lack of reliable experimental techniques for the analysis of the Li-ion distribution[9–16].

Among widely used experimental techniques, scanning transmission electron microscopy (STEM) and advanced X-ray techniques have provided invaluable information, such as the crystalline lattice structure[17–20] and chemical sates[21–30]. Thus far, however, they lack the ability to directly quantify the charge carrier in LIBs, Li ions, at high spatial resolution. Therefore, it is still difficult to comparatively study and explain the capacity decay mechanism of LIBs using the aforementioned techniques and lifetime testing. To achieve this aim, we employed STEM, which reveals the atomic arrangements, together with atom probe tomography (APT) at the same location, which provides 3D quantitative information on the constituent elements, including Li ions, with a sensitivity of ~10 parts per million[31–34].

We report the origin of the irreversible capacity loss of a $Li(Ni_{0.80}Co_{0.15}Mn_{0.05})O_2$ (NCM) cathode material subjected to numerous charge–discharge cycles by correlating its structural evolution with its quantitative 3D composition. We selected an NCM cathode with a high Ni content, which was recently commercialised owing to its high capacity and low material cost[8,10,35]. Its structural and corresponding compositional changes during charge–discharge cycling were investigated to further understand the capacity degradation mechanism. We further demonstrate that the Li concentration is proportional to the amount of Li accommodation sites, which in turn is determined by the structural evolution. Our findings provide a new pathway for addressing the capacity decay in LIBs by understanding the inherent behaviour of Li ions.

## Results

**Capacity fade and the direct observation of Li.** To clarify the correlation between the capacity fade and the loss of Li, we examine the change in the capacity fade with both the global and local Li losses during cycling. Figure 1a reveals that the capacity of the NCM-based full cell (Supplementary Fig. 1) fades continuously to 137.5 mAh g⁻¹, which is 76.0% of the initial specific capacity, after 300 charge–discharge cycles at a C-rate of 1 C. Supplementary

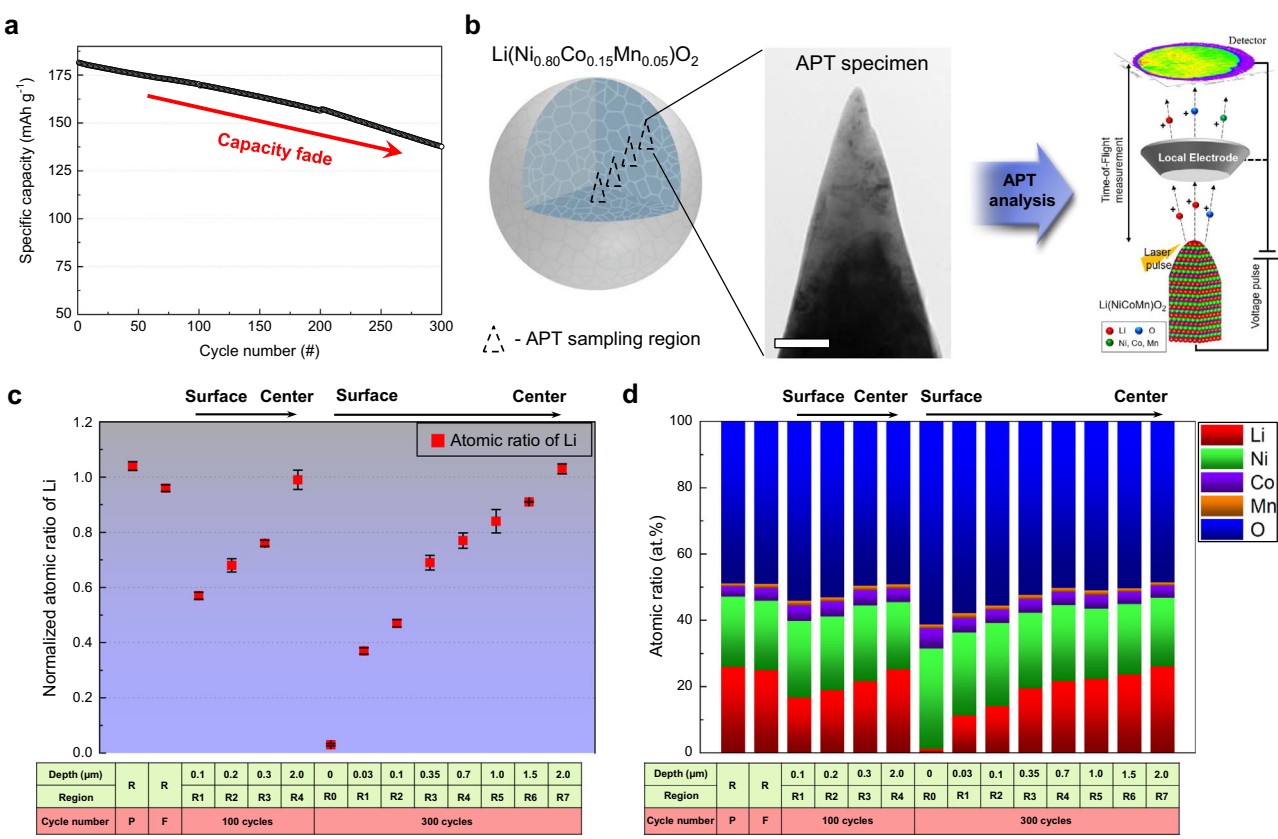

**Fig. 1 Evolution of Li concentration gradient in NCM particle after charge–discharge cycles. a** Specific capacity as a function of the number of cycles, revealing capacity fading in the NCM cell under a C-rate of 1 C at 45 °C. **b** Schematic illustrations of site-specific APT and STEM analysis along the depth of secondary particles to correlate structural degradation with quantitative Li concentration. Scale bar is 200 nm. **c**, **d** Comparison of the normalised atomic ratio of Li (**c**) and compositional changes (**d**), as measured by APT, along the depth of the particles before and after cycling. Pristine and formed NCM are denoted by P and F, respectively. The Li concentration gradient begins to evolve after cycling, and the depth of this gradient expands as the number of cycles increases. Error bars for each data points are from the standard deviation of measured ratios from multiple samples.

**Table 1 Compositions of cationic elements in pristine Li(Ni$_{0.80}$Co$_{0.15}$Mn$_{0.05}$)O$_2$ (NCM) and NCM subjected to 100 and 300 cycles as measured by ICP-AES (the atomic concentration is normalised to obtain a total TM fraction of 1).**

| Specimen | Atomic concentration (weight %) | | | |
|---|---|---|---|---|
| | Li | Ni | Co | Mn |
| NCM-pristine | 1.04 (6.77) | 0.80 (44.31) | 0.15 (8.24) | 0.05 (2.46) |
| NCM-100 cycles | 0.94 (6.16) | 0.80 (44.36) | 0.15 (8.23) | 0.05 (2.46) |
| NCM-300 cycles | 0.83 (5.44) | 0.80 (44.58) | 0.15 (8.29) | 0.05 (2.48) |

**Table 2 Site-specific quantitative APT results along the depth of pristine Li(Ni$_{0.80}$Co$_{0.15}$Mn$_{0.05}$)O$_2$ (NCM) and NCM subjected to the formation, 100 cycles and 300 cycles (the atomic ratio is normalised to obtain a total metal fraction of 1).**

| Specimen | | Atomic ratio | | | | |
|---|---|---|---|---|---|---|
| Process | Region (depth from the surface) | Li | Ni | Co | Mn | O |
| NCM-pristine | R (from surface to centre) | 1.03 | 0.84 | 0.12 | 0.04 | 1.94 |
| NCM formation | R (from surface to centre) | 0.96 | 0.81 | 0.15 | 0.05 | 1.89 |
| NCM-100 cycles | R1 (< 100 nm) | 0.57 | 0.79 | 0.16 | 0.05 | 1.85 |
| | R2 (~200 nm) | 0.68 | 0.79 | 0.16 | 0.05 | 1.90 |
| | R3 (~300 nm) | 0.76 | 0.79 | 0.16 | 0.05 | 1.73 |
| | R4 (~2 μm) | 0.99 | 0.79 | 0.16 | 0.05 | 1.92 |
| NCM-300 cycles | R0 (surface) | 0.03 | 0.81 | 0.16 | 0.03 | 1.63 |
| | R1 (< 30 nm) | 0.37 | 0.81 | 0.14 | 0.05 | 1.88 |
| | R2 (~100 nm) | 0.47 | 0.83 | 0.13 | 0.04 | 1.84 |
| | R3 (~350 nm) | 0.69 | 0.81 | 0.14 | 0.05 | 1.86 |
| | R4 (~700 nm) | 0.77 | 0.82 | 0.14 | 0.04 | 1.79 |
| | R5 (~1.0 μm) | 0.84 | 0.80 | 0.15 | 0.05 | 1.91 |
| | R6 (~1.5 μm) | 0.91 | 0.82 | 0.14 | 0.04 | 1.94 |
| | R7 (~2.5 μm) | 1.03 | 0.82 | 0.14 | 0.04 | 1.91 |

Fig. 2 shows the charge–discharge curves of the cell cycled in the voltage range of 2.8–4.35 V at 0.2 C and 45 °C as a formation process (hereafter, the corresponding cell is referred to as NCM formation). In addition, inductively coupled plasma-atomic emission spectroscopy (ICP-AES) confirms a significant loss of Li from the cathode after battery cycling (Table 1). The atomic ratio of as-prepared NCM (hereafter referred to as NCM-pristine) is Li:Ni:Co:Mn = 1.04:0.80:0.15:0.05, in agreement with the intended stoichiometry. However, the Li concentration decreased dramatically with the number of cycles, whereas the composition of the TMs remained almost the same, as shown in Table 1. The atomic ratio of Li in NCM decreased continuously from 1.04 in NCM-pristine to 0.94 and 0.83 after 100 and 300 cycles of device operation, respectively (the cells subjected to 100 and 300 cycles are hereafter referred to as NCM-100 cycles and NCM-300 cycles, respectively). These results suggest a direct correlation between the amount of Li loss and the corresponding capacity degradation of the battery. However, both the distribution and local quantitative information of Li ions remain unknown.

With this combined approach, we directly observed the Li ions in 3D and quantified the Li loss (Fig. 1b). NCM particles are spherical with diameters of ~10 μm, and each NCM particle consists of hundreds of smaller (200–600 nm) primary particles (Supplementary Figs. 3 and 4). We quantitatively analysed the compositions of NCM-pristine, NCM formation, NCM-100 cycles and NCM-300 cycles as a function of the radial direction from the surface of a secondary particle in contact with the liquid electrolyte, which acts as a pathway for the extraction and insertion of Li ions (Table 2, Fig. 1b–d and Supplementary Fig. 4). Notably, the Li insertion/extraction reaction is active at the particle surface, so the location, quantity and uniformity of the Li remaining in the cathode should be clarified.

In the case of compound materials, however, the accuracy of APT's compositional measurements depends on the analysis conditions such as the laser pulse energy[36–41]. We thus optimised these conditions to accurately quantify the composition without an under/overestimation. Under these conditions, the APT of NCM-pristine revealed the ideal stoichiometry when we applied a laser pulse energy of 25 pJ at 30 K. Its atomic ratio as determined by APT was Li:Ni:Co:Mn:O = 1.03:0.84:0.12:0.04:1.94 (Table 2 and Supplementary Fig. 5), in good agreement with the values from ICP-AES (Table 1). We also used APT to determine the atomic ratio of NCM formation, which was Li:Ni:Co:Mn:O = 0.96:0.81:0.15:0.05:1.89 (Table 2), almost the same as that of NCM-pristine. However, the cycled NCM showed a substantial Li deficiency, with an atomic fraction of Li less than 1.00, except for the most central regions of the particle. Further, the Li content tended to decrease with the increasing number of cycles, similar to the findings from ICP-AES. The APT results hence confirm that the Li content dramatically changed with the depth below the particle surface as well as the number of cycles.

**Evolution/expansion of Li concentration gradient**. The Li distribution in NCM subjected to repeated charge–discharge cycles was remarkably different from those in NCM-pristine and NCM formation (Fig. 1c, d and Supplementary Fig. 6). The APT result of NCM-100 cycles showed a severe Li deficiency at a distance of less than 100 nm from the top surface of the sample (100 cycles–Region 1; for simplicity, the different regions are hereafter referred to as R1, R2, etc., as defined in Table 2). In this region, the atomic ratio determined by APT is Li:Ni:Co:Mn:O = 0.57:0.79:0.16:0.05:1.85, indicating a deviation from those of NCM-pristine and NCM formation (Table 2). Specifically, the atomic fraction of Li decreased significantly from 1.03 for NCM-pristine and 0.96 for NCM formation to 0.57 for R1 in NCM-100 cycles. A Li deficiency was still observed ~200 and ~300 nm below the top surface (100 cycles–R2 and R3, respectively), but it was less pronounced than that of 100 cycles—R1 (Table 2). In contrast, ~2 μm below the surface, the composition was almost the same as that of the pristine sample (Table 2). The results of the quantitative APT analysis thus prove that a significant Li loss occurs at the top surface and that the amount of Li increases with increasing depth from the top surface of the particle, which can be described as a Li concentration gradient along the radial direction of the particle after 100 cycles.

The APT analysis of NCM-300 cycles also demonstrates the evolution of a Li concentration gradient along the radial direction and further reveals that this gradient region expanded. The atomic fraction of Li at the top surface (R0) and at depths of ~30 nm (R1), ~100 nm (R2), ~350 nm (R3), ~700 nm (R4), ~1 μm (R5), ~1.5 μm (R6) and ~2.5 μm (R7) below the top surface are 0.03, 0.37, 0.47, 0.69, 0.77, 0.84, 0.91, 1.03, respectively. Thus, as clearly shown in Fig. 1c and Table 2, the amount of Li gradually increased from the surface to the centre of the particles in both NCM-300 cycles and NCM-100 cycles. However, in NCM-300 cycles, the Li-depleted region extended beyond ~1.5 μm below the surface. In addition, at a similar depth, the amount of Li in NCM-300 cycles is noticeably lower than that in NCM-100 cycles, revealing that the Li concentration gradient extends deeper in NCM-300 cycles owing to an increase in the Li-depleted area. Thus, we suggest that the Li concentration gradient evolves during cycling, and the gradient region increases proportionally to the number of cycles.

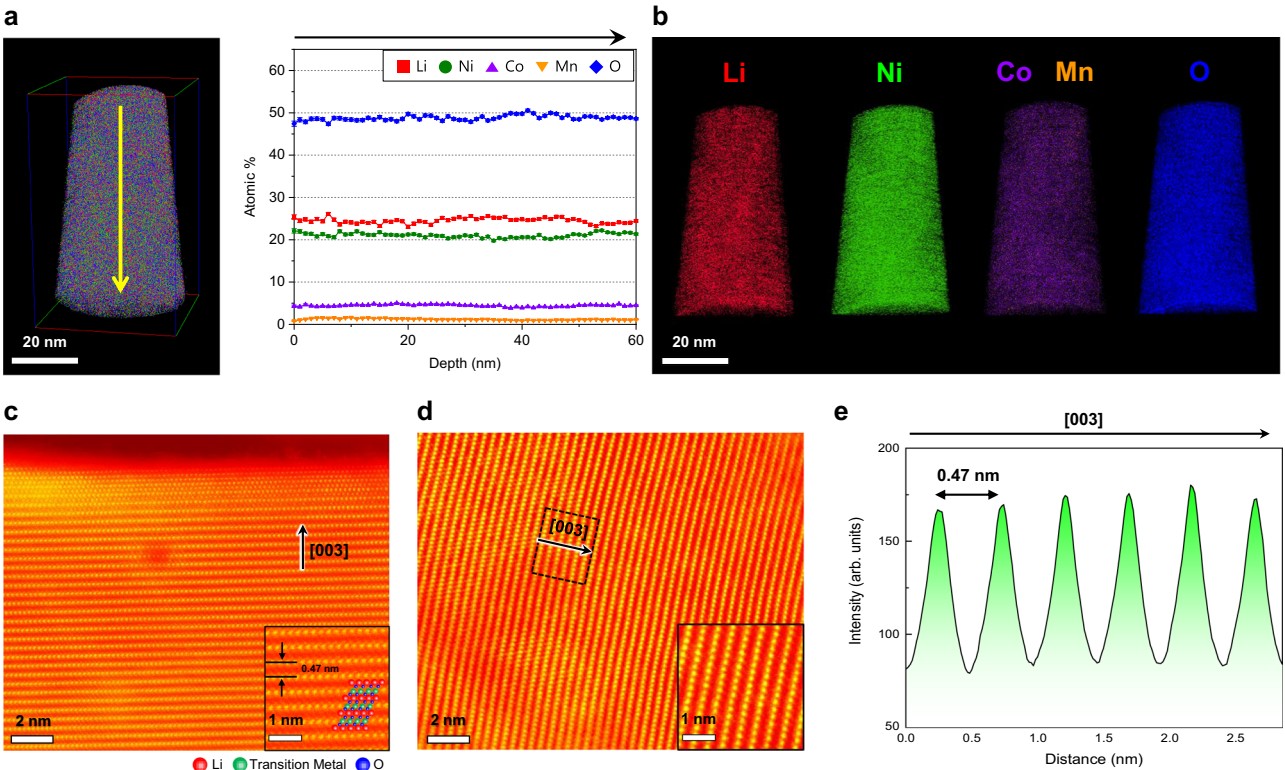

**Fig. 2 Homogenous distribution of Li and the layered structure in NCM-pristine. a** APT depth profile showing a homogeneous distribution of the constituent elements, including Li, along the depth. **b** APT 3D ion map of Li (red), Ni (green), Co (violet), Mn (orange), O (blue), respectively, showing no elemental segregation. Error bars show the standard deviation of each dataset. **c, d** STEM-HAADF images along [1$\bar{1}$0] zone axis. A layered structure is clearly observed both at the surface (**c**) and in the interior (inset: magnified image) (**d**). **e** HAADF intensity profile along the region enclosed by the dotted square in (**d**), which shows the absence of TM migration.

**Structural disordering and Li concentration**. To reveal the origin of the Li concentration gradient, we investigated the microstructural changes associated with the evolution of this gradient by using site-specific STEM and APT analyses. The APT measurements near the surface of NCM-pristine provided atomic ratios that were in good agreement with the data determined by ICP-AES (Fig. 2a). The trends in the Li loss during cycling in the APT and ICP-AES results are also similar, showing a greater Li loss after 300 cycles. Further, as shown in Fig. 2b, Li (red), Ni (green), Co (violet), Mn (orange) and O (blue) were all very uniformly distributed in 3D without any deficiency or segregation. To estimate the homogeneity of the elemental distribution, we performed a frequency distribution analysis. The value of the Pearson coefficient ($\mu$), an indicator of the homogeneity of a distribution, for each constituent element remained low. In addition, the frequency distributions were close to binomial distributions along the depth during cycling, as shown in Supplementary Fig. 7 for Li, Ni and O, which reconfirms that the distribution is quite homogeneous. The STEM–high angle annular dark-field (HAADF) image of NCM-pristine from the identical location also shows a clear layered structure (R-3m) represented by alternating bright TM interlayers and dark Li interlayers, both at the surface of and inside particles (Fig. 2c, d, respectively). The atomic structures of the surface of NCM-pristine and the corresponding bulk structures are superimposed in the inset of Fig. 2c. Note that Li and O atoms could not be detected in the HAADF images because of their low atomic numbers, which is a limitation of the STEM-HAADF technique. The observed regions are indicated by yellow squares in Supplementary Fig. 8. In the intensity profile along the (003) plane, the discrete regions of strong intensity represent the TM layer with a d-spacing of 0.47 nm (Fig. 2e), indicating that TMs rarely migrate into the Li layer.

By contrast, the STEM and APT results of NCM-300 cycles reveal distinct changes in the atomic structures and the corresponding amount of Li as a function of the depth. At the top surface (R0) of NCM-300 cycles, the atomic ratio determined by APT is Li:Ni:Co:Mn:O = 0.03:0.81:0.16:0.03:1.63 (Fig. 3a and Table 2), exhibiting the lowest amount of Li. Below the upper ~15 nm region, the amount of Li gradually increased with the increasing distance from the surface, which implies that this region consists mainly of the TMs and oxygen, along with the most significant Li deficiency (Fig. 3a). The STEM-HAADF image also confirms the evolution of a Li-deficient phase such as a spinel- or rock-salt-like phase induced by the substantial TM migration from octahedral 3a (denoted by 3a) TM sites to octahedral 3b (denoted by 3b) Li sites in the surface region (Fig. 3b and Supplementary Fig. 9). The HAADF intensity profile perpendicular to the (003) plane in the dotted box in Fig. 3b shows a strong intensity between the TM layers, as denoted by red arrows (Fig. 3c). These results directly demonstrate that the disordered structure has a lower capacity to accommodate Li ions than the layered structure.

At ~30 nm from the top surface (R1), the atomic fraction of Li is higher than that at the top surface (R0), which shows the value of 0.37 (Fig. 3d). The depletion of Li in R1 was suppressed by the decrease in the migration of TMs into 3b Li sites. Clearly, the TMs migrated to a lesser degree than they did near the top

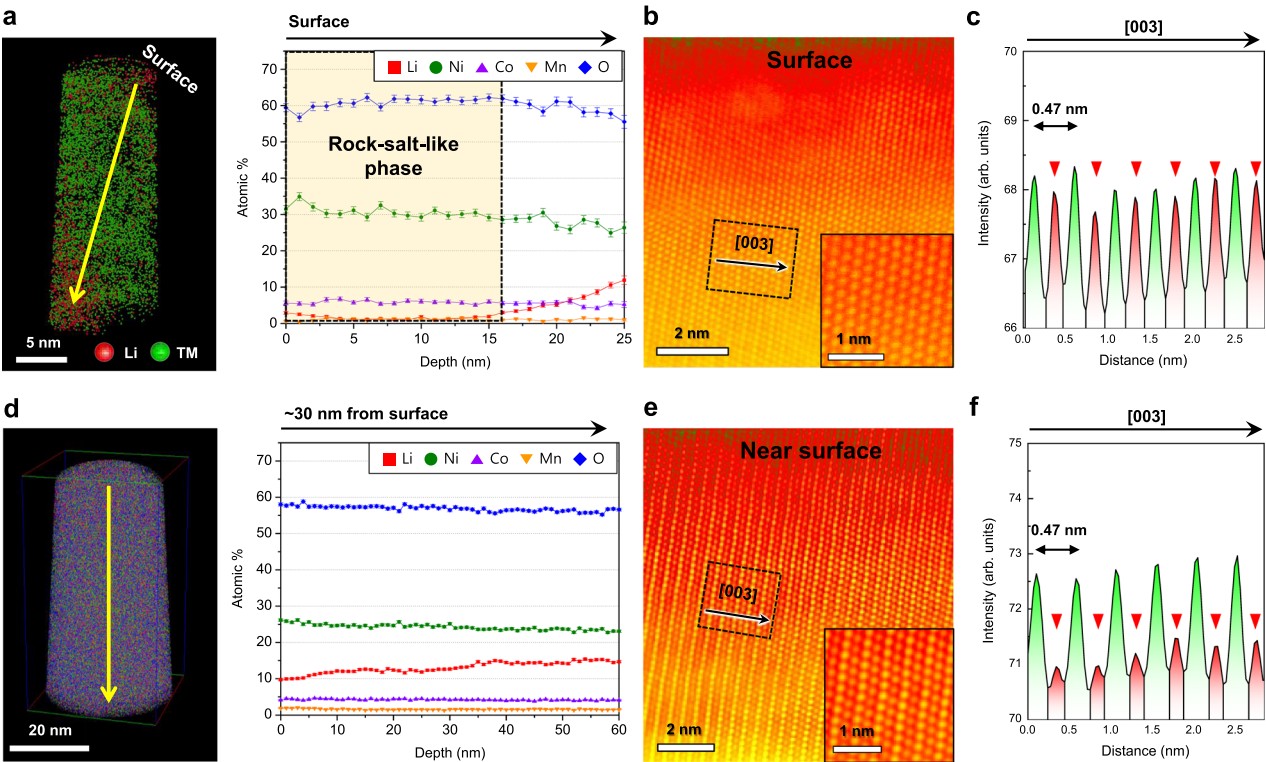

**Fig. 3 Evolution of the Li deficiency and disordered structure near the surface of NCM-300 cycles. a** APT depth profile at the top surface, showing a significant Li deficiency up to ~15 nm (region enclosed by dotted square). **b, c** STEM-HAADF image (**b**) and intensity profile (**c**) at the top surface. The rock-salt-like phase evolves because of TM migration (inset: magnified image). Red arrows indicate TM migration from 3b to 3a sites. **d** APT depth profile starting at ~30 nm from the top surface, showing the gradual increase in the Li concentration along the depth. **e, f** STEM-HAADF image (**e**) and intensity profile (**f**) starting at ~30 nm from the surface (inset: magnified image). The red arrows show that 30 nm below the surface, TMs migrate to a lesser extent than they do near the top surface in (**c**). Error bars show the standard deviation of each dataset.

surface, as denoted by red arrows (Fig. 3e, f). These stark differences in both the Li concentration and the structural disordering at the surface of NCM-300 cycles were mitigated in the interior of the particle. Nevertheless, at ~700 nm from the top surface (R4), the APT depth profile still shows a slightly deficient amount of Li (Fig. 4), but this deficiency is markedly suppressed relative to that near the surface (R0–R3), showing an atomic fraction of Li greater than ~0.70. The partially disordered structure was also formed by just a small amount of TM migration (Fig. 4b). The HAADF intensity profile still reveals the noticeable presence of TMs in the Li layer, but the intensity is significantly weakened, as denoted by red arrows in Fig. 4c. Indeed, the central region of the particle presents almost the same amount of Li as that of NCM-pristine (Fig. 4d). Even after 300 cycles, the layered structure is well maintained in the central region of the particle, with almost no TM migration (Fig. 4e, f). Further, TM migration would not tangibly affect the atomic structure or the Li-ion distribution in the most central region. The observed regions in NCM-300 cycles are indicated by yellow squares in Supplementary Fig. 10. At low magnification, crack formation or an increased gap between primary particles was also observed after 300 cycles, as previously reported many times (Supplementary Fig. 10)[42–45]. Our site-specific combined STEM and APT analysis unequivocally confirm that the evolution of the Li concentration gradient after cycling could be attributed to structural disordering. More specifically, the capacity to accommodate Li ions depends on the degree of structural disordering. Therefore, the Li concentration gradually increases from the surface to the centre of the particle as the degree of structural disordering decreases.

## Discussion

**Li accommodation in the layered structure.** Our site-specific APT results reveal the development of an increasing Li concentration gradient along the depth of the NCM particles and signify the expansion of the gradient region as cycling proceeds. Through the systematic STEM and APT observations, we further discovered that the evolution/expansion of the Li concentration gradient was associated with structural disordering. While the Li ions move toward the anode during the charging process, TMs readily move to and occupy the 3b Li sites in the Li layer via the tetrahedral sites because the formation of a disordered structure is energetically favourable in the O3-type layered structures (Fig. 5a)[14,46–48]. The TMs that occupy the 3b Li sites are thermodynamically stable in the Li-deficient phase because of the elimination of the repulsive forces from neighbouring cationic species, which would normally facilitate TM migration. In contrast, the Li ions hardly occupy the 3a TM sites, whereas TMs occupy the 3b Li sites during discharge, as shown schematically for the partially disordered structure (Fig. 5a). The inserted Li ions simply refill a few remaining vacant 3b Li sites, thereby avoiding refilling the vacant 3a TM sites. Concurrent with the preference of TMs to migrate to Li sites, the prevention of Li migration to TM sites leads to the loss of Li accommodation sites for the inserted Li ions during the discharge process. The number of Li accommodation sites varies with the degree of structural disorder.

The particle surface in contact with the electrolyte represented as the rock-salt- or spinel-like disordered phase has a significantly lower number of Li accommodation sites. The interior of the particle (at a depth of 0.1–1.5 μm from the

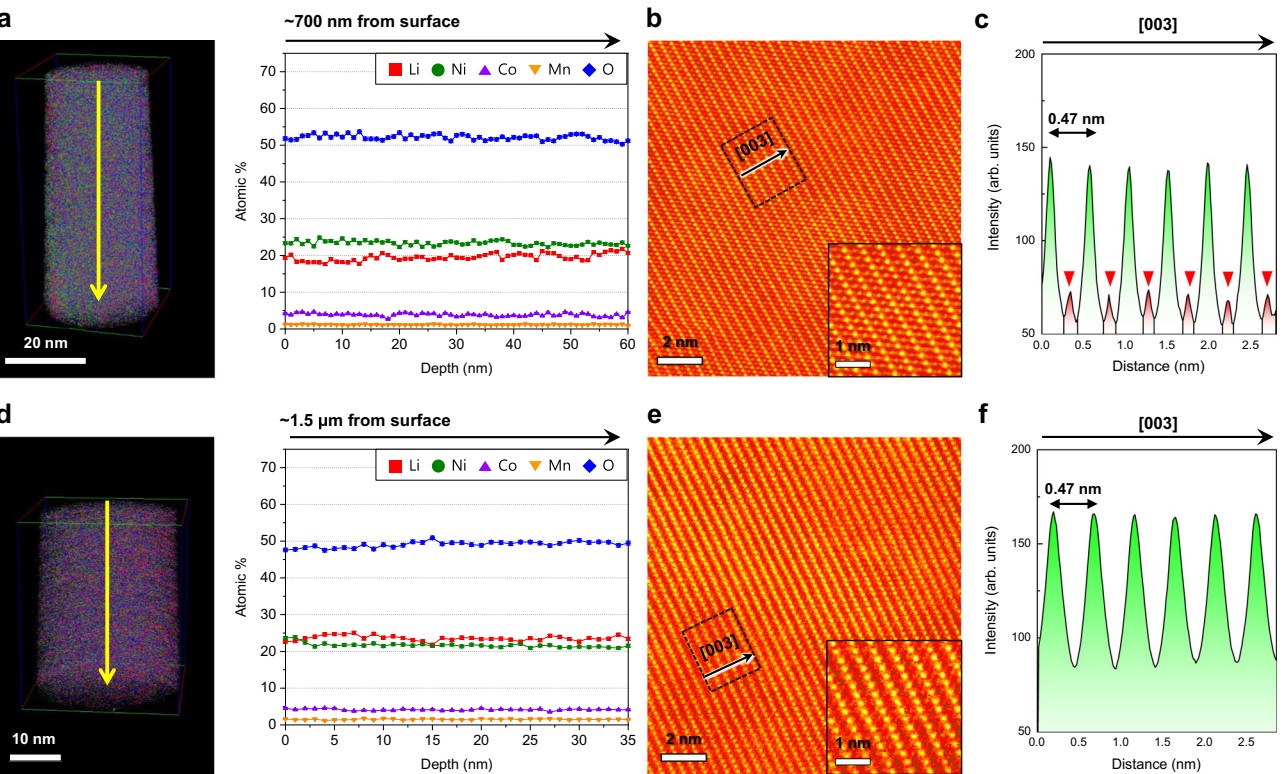

**Fig. 4 Mitigation of Li deficiency and disordering in the interior of NCM-300 cycles. a** APT depth profile at ~700 nm from the top surface, showing slightly deficient Li concentration. **b**, **c** STEM-HAADF image (**b**) and intensity profile (**c**) sampled at depths of 500 nm – 1 μm. A partially disordered structure (inset: magnified image) formed owing to the low degree of TM migration to 3a sites (red arrows) **d** APT depth profile in the central region of the NCM particle, showing the same Li concentration as that of NCM-pristine. **e**, **f** STEM-HAADF image (**e**) and intensity profile (**f**) in the central region. Layered structure (inset: magnified image) is well maintained, showing no TM migration. Error bars show the standard deviation of each dataset.

surface) represented by a partially disordered structure also has fewer Li accommodation sites. The centre of the particle, which retains a layered structure, has the original number of Li accommodation sites. Because more Li ions preferentially occupy the vacant Li accommodation sites, the Li concentration gradient from the surface to the centre of the particle evolves/ expands after cycling (Fig. 5b). If the inserted Li ions occupy both the vacant Li and TM sites, the original Li concentration should have been preserved even in the disordered structure. Hence, according to our APT and STEM results, it is reasonable that the site preference of Li ions according to the degree of structural disorder is the dominant factor that determines the total number of Li accommodation sites and consequently the Li concentration. Considering that the structural disorder correlates with the Li loss, our results provide direct evidence for the irreversible migration of TMs in the O3-type layered structures.

**Non-homogeneous distribution of Li**. According to our APT analysis, most of the regions in NCM show homogeneous elemental distributions with no segregation (Figs. 3 and 4 and Supplementary Fig. 11). However, we also observed the non-uniform distribution of constituent elements at several sites after cycling, in agreement with previous reports on local compositional variations[16,33,34]. The values of μ for Li are higher than 0.9, and the observed frequency distributions significantly deviate from binomial distributions, indicating a higher degree of non-uniformity, as shown in Supplementary Fig. 12. Figure 6a and b

shows the existence of regions locally enriched with Li, which mainly follow a line shape. Interestingly, this line-shaped inhomogeneous Li distribution is consistent with the general shape of crystalline line defects, such as dislocations and anti-phase domain boundaries, as observed in the STEM-HAADF images (Fig. 6c, d). Although this study does not clarify the detailed relationship between the arrangement of atomic defects and Li accommodation sites, we could conclude that these defects change the local atomic arrangement and hence the lattice strain[49–51], and eventually the distribution of Li[51]. The defects, which serve to concentrate strain, change the stacking sequence and stabilise the coexistence of O3- and O1-type layered structures via the formation of regions locally enriched with Li. However, owing to the extremely low fraction of the defective volume relative to that of the entire cathode, the capacity degradation is mainly caused by the Li deficiency represented by the Li concentration gradient.

In summary, we analysed the site-specific composition of the NCM cathode material, including the Li-ion distribution, via APT. The Li concentration gradient along the radial direction in secondary particles, which decreased from a certain depth toward the particle surface, evolved after charge–discharge cycles. The depth of the gradient expanded, and the extent of Li depletion increased as the number of cycles increased, resulting in the capacity fade of LIBs. Our complementary analysis using identical-location STEM and APT reveals that the local Li deficiency originates from the lack of Li accommodation sites due to the migration of TM elements. The vacant 3a TM sites do not

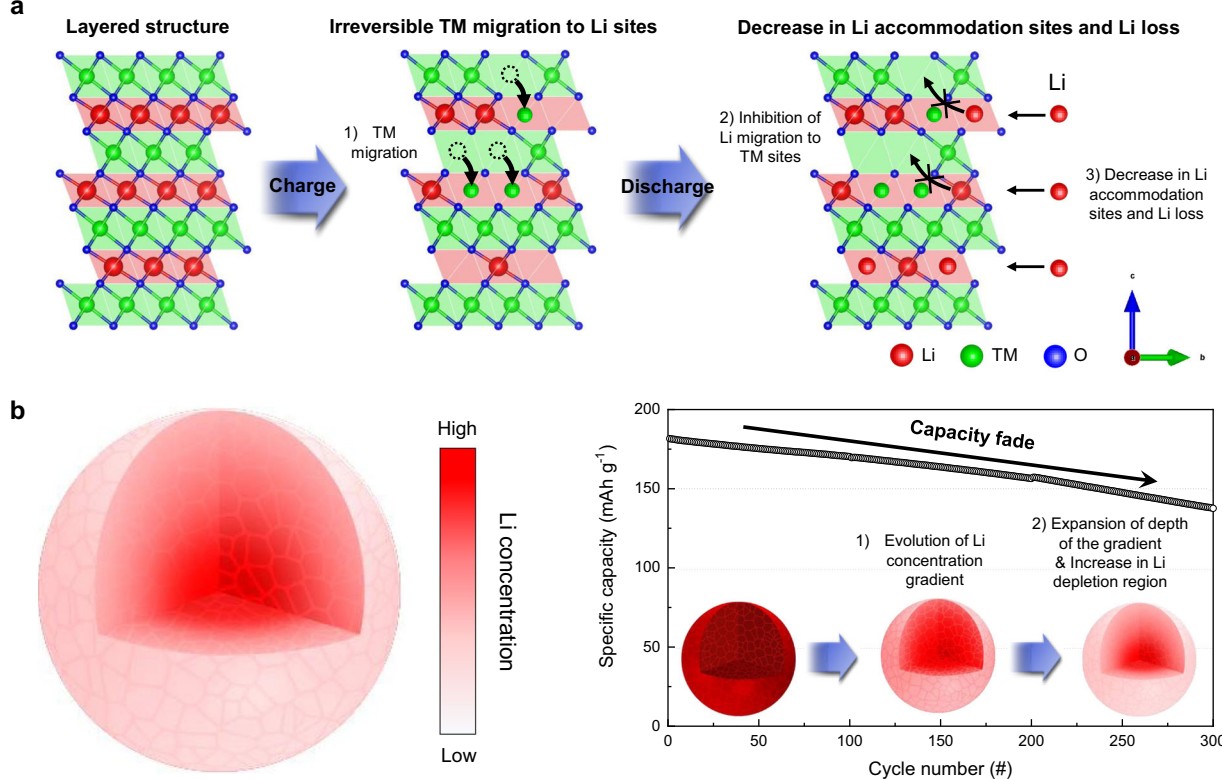

**Fig. 5 Comparison of Li distribution with the increasing number of cycles. a** Schematic illustration of Li accommodation sites during the charge–discharge process. TMs occupy 3b Li sites in the Li layer during the charging process. This irreversible TM migration depletes the number of Li accommodation sites in the Li layer. In contrast, Li ions simply refill a few vacant 3b Li sites, avoiding the 3a TM sites during the discharge process. Thus, a local Li deficiency originates from the lack of Li accommodation sites, depleted by the irreversible TM migration. **b** Schematic illustrations of the Li distribution evolution with the increasing number of cycles. The Li concentration gradient along the radial direction, which is lower at the surface and higher in the central area, evolves during cycling. The depth of the gradient and the extent of Li depletion both increase with the increasing number of cycles owing to the lack of Li accommodation sites.

seem to accommodate Li. Our analyses suggest that a promising strategy for improving the cycle life characteristics of LIBs is to suppress the decrease in Li reservoir sites during battery operation.

## Methods

**Synthesis of NCM**. The cathode material was synthesised by a solid-state reaction. LiOH·$H_2$O and co-precipitated hydroxide (($Ni_{0.80}Co_{0.15}Mn_{0.05}$)(OH)$_2$) were used as the starting materials. To compensate for lithium loss during heat treatment, a slightly higher ratio of lithium was used with respect to the mixed metal hydroxide (Li/(Ni+Co+Mn) = 1.03). The starting materials were mixed in a Hansel mixer at 1000 rpm for 15 min. Subsequently, the mixture (150 g) was heated at 750 °C under an air stream (20 L/min) for 8 h.

**Electrochemical measurements**. The electrochemical properties of the synthesised materials were evaluated with the use of cylindrical 18,650 cells (Supplementary Fig. 1). Artificial graphite was used as the anode. For preparing the cathode, the synthesised powder was mixed with acetylene black and a PVDF binder in 92:4:4 ratio (wt.%). The loading level of the cathode was 17.7 ± 0.1 mg/$cm^2$. A solution of 1.15 M $LiPF_6$ in a mixture of ethylene carbonate/ethyl methyl carbonate/dimethyl carbonate (1:1:3 vol.%) was used as the electrolyte. For the activation of the cells, a charge–discharge cycle of the cylindrical cell at 0.2 C was performed twice between 4.35 and 2.8 V at 45 °C, and the capacity retentions were subsequently recorded at 1.0 C and a temperature of 45 °C in the same voltage range for 300 cycles. All cells were disassembled in a humidity-controlled room after being discharged.

**STEM imaging**. The cross-sectional STEM specimens were fabricated with the use of a focused ion beam (FIB, Helios Nano-Lab 600). To minimise the damage caused by the Ga-ion beam, final milling was performed using a voltage of 5 kV

and 2 kV at currents of 16 pA and 3 pA, respectively. The microstructure of the NCM particles was characterised by CS-corrected STEM (FEI-Titan G2 and Titan Cubed). To better visualise the structure, the region of interest is magnified and cropped (Supplementary Fig. 13).

**APT specimen preparation**. The needle-shaped specimens for APT were fabricated by a FIB lift-out method. On the top surface of the NCM particles, a 100-nm-thick Ag layer was deposited by sputtering at room temperature for passivation. The NCM particles were dispersed on a piece of carbon tape attached to a Si substrate. For additional passivation, a 12 × 2 μm region of interest on the particle surface was deposited with 100 nm thick Pt and 1-μm-thick Pt layers using an electron beam and a Ga-ion beam in FIB, respectively. The passivated region of interest was lifted off and transferred onto a pre-sharpened W tip. These samples were sharpened in an annular pattern using a voltage of 30 kV at a current of 93 pA. To resharpen the samples and minimise the Ga-ion beam damage, final milling was performed using a current of 5 and 2 kV at currents of 16 and 3 pA, respectively. To compare the composition as a function of the depth from the particle surface using APT, the distance of the analysed location from the particle surface was measured in the FIB electron beam image.

**APT analysis**. APT analysis was performed using a CAMECA local electrode atom probe (LEAP) 4000X HR in the UV laser mode ($\lambda$ = 355 nm) at a 125 kHz pulse repetition rate and 40 K base temperature. To avoid the effect of the APT analysis conditions on the measured chemical composition of NCM, we used laser-pulsed energy of 25 pJ at 0.005 atoms/pulse detection rate. Five APT specimens were analysed at each depth in the NCM particles to verify the concentration gradient. APT reconstruction and data processing were carried out using CAMECA Integrated Visualisation and Analysis (IVAS) software. SEM and TEM images of the tips were used for APT reconstructions. The APT mass spectrum of NCM is

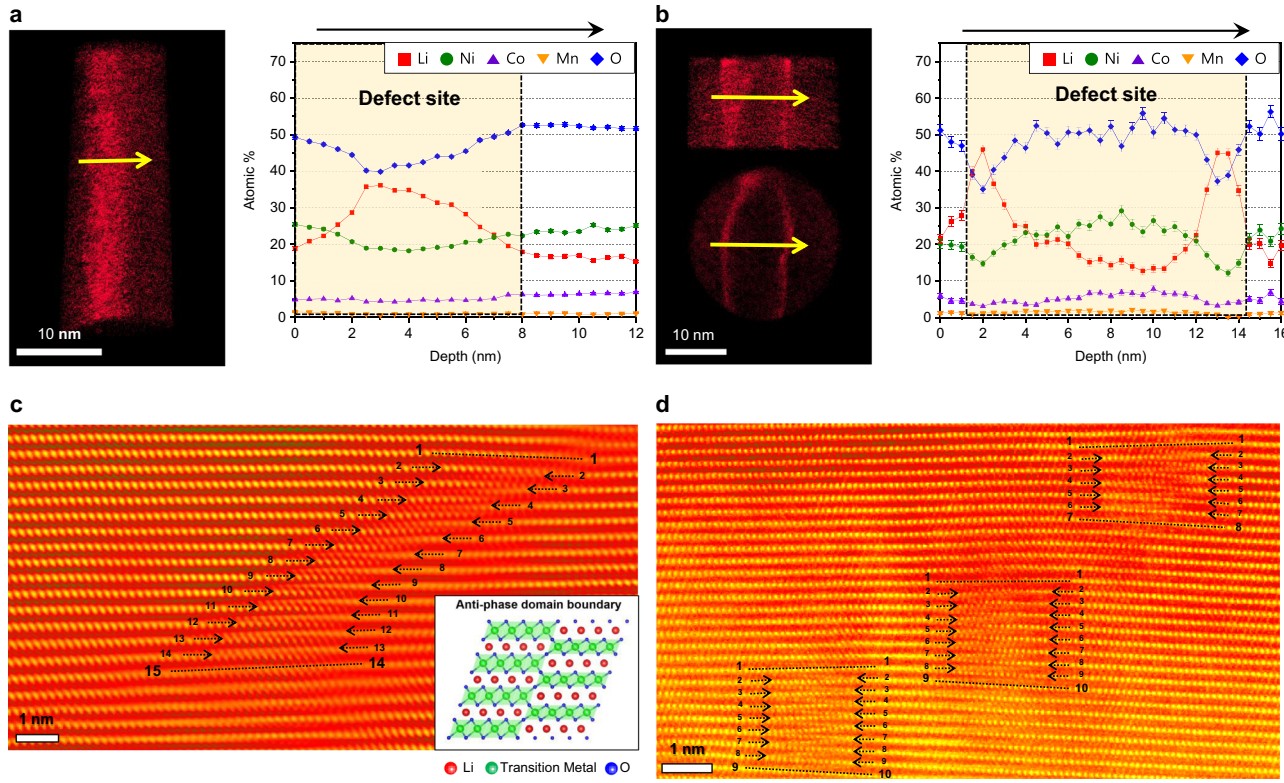

**Fig. 6 Non-homogeneous distribution of Li and formation along line-shaped defects after charge–discharge cycles. a, b** APT Li map and composition profile across line-shaped Li-enriched regions. The Li concentration is locally high, following a line shape. Error bars show the standard deviation of each dataset. **c, d** STEM-HAADF images of the anti-phase domain boundary (APDB) (**c**) and the arrangement of APDB in the grain (**d**). Inset in (**c**) is a schematic illustration of APDB.

exhibited in Supplementary Fig. 14. A pole is not visible in our APT data, as shown in Supplementary Fig. 15.

## Data availability

The data that support the findings of this study are available from the corresponding author upon reasonable request.

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

## Acknowledgements

We thank Seongwoo Hwang, the former corporate president of SAIT, for helpful discussions.

## Author contributions

E.L., W.S.J., and S.Y.P. conceived the research plan. J.H.S. synthesised Li $(Ni_{0.80}Co_{0.15}Mn_{0.05})O_2$ and tested cells. B.G.C. conducted the APT study. S.Y.P. and B.G. C. conducted TEM. B.G.C. and S.Y.P. wrote the paper. All authors discussed the results and commented on the manuscript.

## Competing interests

The authors declare no competing interests.
