## [Peer Review File · Nature Communications]

Reviewer #1 (Remarks to the Author):

A combined APT/STEM approach was used to identify the origin of capacity loss after several charge/discharge cycles from a Ni-rich layered oxide cathode (NMC). Li concentrations were measured using APT at different depths in the material, ranging from within nms of the surface to 2.5 μm from the surface. The results showed that a Li concentration gradient with respect to depth from the surface after cycling. A Li depletion was found at distances closer to the surface and stoichiometric Li concentrations were found toward the center of the particles. STEM-HAADF images corroborated these results showing that transition metals (TMs) were infiltrating Li sites in the structure causing disorder and eventual phase changes. The TMs fill the vacant Li sites during charging decreasing the number of Li accommodation sites in the cathode, which decreases the cathode capacity. Overall, the paper is well written, easy to understand, is scientifically accurate, and has a very coherent and strong scientific story. I highly recommend this manuscript for publication in Nature Communications after addressing my very few minor comments. I have also attached a word file with tracked changes to show some minor comments and grammatical changes.

1. For Fig. 1, it would be a lot easier to understand if the distances from the surface were labeled instead of having R1, R2, etc. It is difficult to fully comprehend the data when referencing Table 2. I also suggest making another figure with Normalized Li concentration vs. approximate distance from the surface and use different fiducials for the 100 and 300 cycled samples. That way the results can be more directly compared.
2. The section with the heading, "Li accommodation in the layered structure" should be the start of the discussion section. The authors should have the heading Discussion followed by the sub-heading "Li accommodation in the layered structure." The section marked Discussion in the manuscript should be labeled Conclusions.
3. See the attached Word file for minor edits.

Jonathan D. Poplawsky

Reviewer #2 (Remarks to the Author):

The authors report on an interesting approach to reveal the mechanism underpinning the loss of capacity of Li-ion batteries by investigating the Li concentration and depth distribution as a function of cycling by a combination of APT and STEM.

Overall I think the article is convincing, well written, clearly an advance on the state-of-the-art, and should be published. The mechanisms unveiled may not have been unforeseen or unexpected but they are now proven beyond doubt, and that is essential.

I would however like to point a few issues that would be important to address:

1. first the use of correlative microscopy around APT/STEM is typically used to refer to the use of TEM-based techniques performed directly on the APT specimen - see

<https://www.sciencedirect.com/science/article/pii/S1359646217301422>

I think this should be made clear to the reader that it is not the case here. The approach used here can be referred to as "identical location" for instance - see

<https://www.sciencedirect.com/science/article/pii/S0360319917333578>

I think this would be better and more accurate.

2. wrt the composition reported from APT would be good to add errorbars or error estimation. There are aspects of the discussion that would benefit from this. This is particularly the case for the discussion on possible Li-losses – we know APT is not so good with oxygen:) – but for Li not much has been done. This would be a good resource for future studies. Please see detailed comments and references in the attached document. I think there might also be an issue with some of the references in the submitted manuscript.

3. it is not completely clear how the authors estimated the depth at which the APT analysis was performed, or how accurate this was, yet this is important for the discussion.

4. I would recommend that the authors dig a bit deeper into their APT data. First at this stage they report on visual inspections of the data to assess the homogeneity of the distribution of Li and other species, which is not good enough IMHO unless it can be justified. Second, if the orientation of the sample is known, even approximately, it is likely that atomic planes can be imaged or searched for from within the data. Could be that a pole is visible? if the authors have trouble with this or making this clear they could seek support or help from other experts in the community (I would be happy to help search if necessary). This might provide additional information that could be valuable.

5. I think it is unnecessary to justify using APT in current times - especially not "against" STEM but to complement STEM. I commented on this in the attachment.

I have made some more detailed comments on the written text in the attached doc file - typos etc.

In any case, the authors are commended for a beautiful in-depth study!

Dr Baptiste Gault

Reviewer #3 (Remarks to the Author):

Manuscript#: NCOMMS-20-50711 Review

Comments for Authors

This manuscript provides scanning transmission electron microscopy (STEM) and atom probe tomography (APT) data on $\text{Li}(\text{Ni}_{0.80}\text{Co}_{0.15}\text{Mn}_{0.05})\text{O}_2$ (NCM) oxides. The samples are from a pristine electrode and from oxide electrodes from cells that underwent 100 and 300 cycles in NMC/Gr cylindrical 18650 cells. The authors correlate changes in the local Li composition in the oxide (obtained by APT) and the structural changes (obtained by STEM) to cell capacity degradation on cycling. The paper is written well and relatively easy to read.

The use of STEM to examine structural changes on cycling is not novel and has been used for over a decade; reference 10 in the manuscript gives one such example. The finding of rock-salt structure on the oxide surface is also not new; such data were reported by Abraham et al. [Electrochemistry Communications 4 (2002) 620–625] about 18 years ago.

The use of APT to study oxide degradation on cycling can be considered novel. APT has been used previously to study other battery oxides, e.g. [<https://doi.org/10.1038/ncomms9014>]. Although the

technique is interesting, interpretation of the results is difficult because of artefacts that could be introduced during the sample preparation/measurement process. Here, the authors appear to have done their best to “calibrate” the data, based on results obtained on the pristine sample. However, in the opinion of this reviewer, the data appear to be inconsistent and, in some ways, incomplete.

Some questions and concerns about the manuscript are as follows:

About the Experimental –

- How many formation “cycles” were conducted? Only 1 cycle is shown in Figure S2.
- What is the thickness of the oxide electrodes used in the study? From which part of the electrode were the particles selected for study? From the electrode surface? Oxide particles closest to the electrode surface are expected to have a higher degree of degradation than those closer to the current collector. See Gilbert et al. [Adv. Mater. Interfaces 2018, 5, 1701447] and Yang et al. [Adv. Energy Mater., vol. 9, no. 25, p. 1900674, Jul. 2019.]
- Were the oxide electrodes “harvested” with the cell in the discharged state? Were these electrodes washed before APT and STEM sample preparation?
- Were the APT samples prepared in air? If so, could there be Li loss from the surface regions because of reaction with the ambient? These reactions would be pronounced for the delithiated oxides that are extracted from the cycled electrodes.
- What is the probe size for the APT experiments? Is the data from one primary particle or several primary particles? The authors indicate that these primary particles are 200-600 nm in size.

About the Results –

- The authors appear to correlate capacity fade of the cell with changes in the local Li composition of the oxide, which is incorrect. Capacity fade of a NMC/Gr cell results from loss of mobile Li-ions to the solid electrolyte interphase (SEI) of the graphite electrode. These “lost” ions are from the oxide and hence the oxide becomes more Li-deficient on cycling, as the authors have shown. Loss of mobile Li-ions is the “definition” of capacity fade. However, this loss is due to electrolyte reduction reactions at the graphite electrode.
- The NCM particles studied here are spherical with diameters of $\sim 10 \mu\text{m}$. The APT data are obtained up to a depth of 2.5 microns from the secondary particle surface. In both the 100-cycle and 300-cycle samples, below a depth of 2.5 microns, the oxide concentration is similar to that of the pristine

oxide (Lines 128-129; 136-137). Because the oxide particles would have a 5 micron radii, these APT data would suggest that the particles between 2.5 and 5 microns are essentially inactive. Why? Because, once extracted, it is hard to put the Li-ions back into the oxide particles – this phenomena described as an “apparent” first cycle irreversibility has been described by Dahn et al [J Electrochem. Soc 147:3598 (2000) & Kang et al. [Mater Sci (2008) 43:4701–4706]. That is, the APT data suggest inactive particles in the oxide, which is incorrect because all particle appear to participate in the electrochemical reaction (as suggested by Figure S2).

- **Lines 135-138** state the following: “The Li concentration at the top surface (R0) and at depths of ~30 nm (R1), ~100 nm (R2), ~350 nm (R3), ~700 nm (R4), ~1 μ m (R5), ~1.5 μ m (R6), and ~2.5 μ m (R7) below the top surface are 0.03, 0.37, 0.47, 0.69, 0.77, 0.84, 0.91, 1.03, respectively. Thus, as clearly shown in Fig. 1c and Table 2, the Li concentration gradually increased from the surface to the center of the particles.” These data indicate that the Li sites in oxide particles close to the interior are full while those close to the exterior are empty. That is, there are high-impedance particles at the secondary particle surface and low-impedance particles close to the interior. If this is the case, how do the Li-ions reach the secondary particle interior? Are all secondary particle “cracked”, so that the electrolyte can penetrate into the interior?
- The Li concentration in Table 2 has been normalized so that the total metal fraction is 1. The proportion of Ni, Co and Mn are kept the same. How does the APT data account for the transition metal dissolution that happens from the oxide?
- About the STEM results – The scale bar in the STEM figures is 2 nm; the data shown is from an area that is about 10 nm x 10 nm. Note here that the primary particle size is 200 – 600 nm. If all primary particles at the surface of the 10 micron size secondary particle have very little Li, then Figure 3 (300 cycle oxide) would show that every portion of the oxide particle has the rock-salt structure: not just the 10 top nm, but the entire 200 nm would be rock-salt.
- The point here is that the STEM data and the APT data are on different scales: the STEM results are on a nanometer scale, whereas the APT data are on a micron scale. It may not be appropriate to correlate (& draw broad conclusions from) the data because they provide different types of information, which are complementary. This is the problem with the schematic in Figure 5. The atomic-scale STEM data is used to justify the micron-scale APT data.

About the Discussion –

This paragraph is simply a restatement of the results and not an explanation/discussion of the data.

In summary, the STEM data shown are reasonable and consistent with similar information shown in the research literature. The electrochemistry data and the SEM images are also ok. However, the APT data may contain artefacts that distort the data. It is likely that the results will not be reproducible. Moreover, studies on a few oxide particles are not representative of a cycled electrode because of the inhomogeneity that results from the lithium extraction/insertion cycling. Hence, the conclusions, as depicted in the schematic of Figure 5, are likely incorrect.

Reviewer COMMENTS:

Reviewer #1 (Remarks to the Author):

A combined APT/STEM approach was used to identify the origin of capacity loss after several charge/discharge cycles from a Ni-rich layered oxide cathode (NMC). Li concentrations were measured using APT at different depths in the material, ranging from within nms of the surface to 2.5 μm from the surface. The results showed that a Li concentration gradient with respect to depth from the surface after cycling. A Li depletion was found at distances closer to the surface and stoichiometric Li concentrations were found toward the center of the particles. STEM-HAADF images corroborated these results showing that transition metals (TMs) were infiltrating Li sites in the structure causing disorder and eventual phase changes. The TMs fill the vacant Li sites during charging decreasing the number of Li accommodation sites in the cathode, which decreases the cathode capacity. Overall, the paper is well written, easy to understand, is scientifically accurate, and has a very coherent and strong scientific story. I highly recommend this manuscript for publication in Nature Communications after addressing my very few minor comments. I have also attached a word file with tracked changes to show some minor comments and grammatical changes.

Author reply to comment

We appreciate for the reviewer's deep understanding about our manuscript and some valuable comments to improve the quality of the manuscript.

1. For Fig. 1, it would be a lot easier to understand if the distances from the surface were labeled instead of having R1, R2, etc. It is difficult to fully comprehend the data when referencing Table 2. I also suggest making another figure with Normalized Li concentration vs. approximate distance from the surface and use different fiducials for the 100 and 300 cycled samples. That way the results can be more directly compared.

Author reply to comment

We thank the reviewer for the helpful comment. Per this suggestion, we labelled the distances from the surface in Fig. 1c and d for better understanding. In addition, we added the normalised atomic ratio of Li vs the approximate distance from the surface graph by using different symbols in Supplementary Fig. 6.

The following changes were made:

Please see Fig. 1c and d,

Please see Supplementary Fig. 6,

Supplementary Fig. 6. Normalised atomic ratio of Li measured by APT along the depth of the particles before and after 100 and 300 cycles. For direct comparison, the atomic ratio of each specimen was normalised and is expressed as a function of the approximate distance from the surface

2. The section with the heading, “Li accommodation in the layered structure” should be the start of the discussion section. The authors should have the heading Discussion followed by the sub-heading “Li accommodation in the layered structure.” The section marked Discussion in the manuscript should be

labeled Conclusions.

Author reply to comment

We thank the reviewer for these helpful comments and fully agree with the reviewer's advice. Per this suggestion, we added headings to the discussion and conclusions sections.

The following changes were made:

Please see Page #9,

“Discussion

Li accommodation in the layered structure”

Please see Page #12,

“Conclusions”

3. See the attached Word file for minor edits.

Author reply to comment

We appreciate these helpful comments and fully agree with the reviewer's advice. Per these suggestions, we corrected the following grammatical errors and implemented these minor edits.

The following changes were made:

Please see Page #3,

“However, these mechanisms were primarily proposed based on the observation of the positions and chemical states of the transition metals (TMs) and oxygen atoms, but not those of Li ions, which are directly responsible for the battery operation **because of** the lack of reliable experimental techniques for the analysis of the Li ion distribution^{9–16}.”

“To achieve this aim, we employed STEM, which reveals the atomic arrangements, together with atom probe tomography (APT), which provides 3D quantitative information on the constituent elements, including Li ions, with a sensitivity of **~10 parts per million**^{31–34}.”

Please see Page #4,

“In addition, inductively coupled plasma-atomic emission spectroscopy (ICP-AES) confirms a significant loss of Li from the cathode **after battery cycling (Table 1)**.”

Please see Page #6,

“The APT result of NCM-100 cycles showed a severe Li deficiency at a distance of less than 100 nm from the top surface of the sample (100 cycles–Region 1; for simplicity, the different regions are hereafter referred to as R1, R2, etc., as defined in **Table 2**).”

“The results of the quantitative APT analysis thus prove that a significant Li loss occurs at the top surface and that **the amount of Li increases with increasing depth** from the top surface of the particle, which can be described as a Li concentration gradient along the radial direction of the particle after 100 cycles.”

Please see Page #7,

“However, in NCM-300 cycles, **the Li-depleted region extended** beyond ~1.5 μm below the surface.”

Please see Page #8,

“Note that Li and O atoms could not be detected in the HAADF images because of their low atomic numbers, which is a limitation of **the STEM-HAADF technique**.”

“In the intensity profile along the (003) plane, the discrete regions of strong intensity represent the TM layer with a d-spacing of 0.47 nm (Fig. 2e), **indicating that TMs rarely migrate into the Li layer**.”

Please see Page #8,

“At ~30 nm from the top surface (R1), the atomic fraction of Li is higher than that at the top surface (R0), which shows the value of 0.37 (Fig. 3d).”

Please see Page #9,

“Nevertheless, at ~700 nm from the top surface (R4), the APT depth profile still shows a slightly deficient amount of Li (Fig. 4), but this deficiency is markedly suppressed relative to that near the surface (R0–R3), showing an atomic fraction of Li greater than ~0.70.”

Please see Page #11,

“Although this study does not clarify the detailed relationship between the arrangement of atomic defects and Li accommodation sites, we could conclude that these defects change the local atomic arrangement and hence the lattice strain⁴⁹⁻⁵¹, and eventually the distribution of Li⁵¹.”

Please see Page #12,

“However, owing to the extremely low fraction of the defective volume relative to that of the entire cathode, the capacity degradation is mainly caused by the Li deficiency represented by the Li concentration gradient.”

Please see Page #14,

“APT analysis was performed using a CAMECA local electrode atom probe (LEAP) 4000X HR in the UV laser mode ($\lambda = 355$ nm) at a 125 kHz pulse repetition rate and 40 K base temperature.”

Reviewer #2 (Remarks to the Author):

The authors report on an interesting approach to reveal the mechanism underpinning the loss of capacity of Li-ion batteries by investigating the Li concentration and depth distribution as a function of cycling by a combination of APT and STEM. Overall I think the article is convincing, well written, clearly an advance on the state-of-the-art, and should be published. The mechanisms unveiled may not have been unforeseen or unexpected but they are now proven beyond doubt, and that is essential. I would however like to point a few issues that would be important to address:

Author reply to comment

Overall, we appreciate the reviewer's valuable and very specific comments. According to each comment, we checked the data again, and we believe that the quality of this manuscript has been significantly improved by these revisions. The corrections corresponding to each comment are as follows.

1. First the use of correlative microscopy around APT/STEM is typically used to refer to the use of TEM-based techniques performed directly on the APT specimen - see

<https://www.sciencedirect.com/science/article/pii/S1359646217301422>

I think this should be made clear to the reader that it is not the case here. The approach used here can be referred to as "identical location" for instance - see

<https://www.sciencedirect.com/science/article/pii/S0360319917333578>

I think this would be better and more accurate.

Author reply to comment

We thank the reviewer for this helpful comment on the use of academically correct terminology. We corrected the relevant terms to avoid confusion.

The following changes were made:

Please see Page #2,

“Here, we present the complementary use of scanning transmission electron microscopy and atom probe tomography at identical locations to demonstrate that the evolution of the local Li composition and the corresponding structural changes at the atomic scale cause the capacity degradation of $\text{Li}(\text{Ni}_{0.80}\text{Co}_{0.15}\text{Mn}_{0.05})\text{O}_2$ (NCM), an LIB cathode.”

Please see Page #3,

“To achieve this aim, we employed STEM, which reveals the atomic arrangements, together with atom probe tomography (APT) at the same location, which provides 3D quantitative information on the constituent elements, including Li ions, with a sensitivity of ~10 parts per million³¹⁻³⁴.”

Please see Page #12,

“Our complementary analysis using identical-location STEM and APT reveals that the local Li deficiency originates from the lack of Li accommodation sites due to the migration of TM elements.”

2. wrt the composition reported from APT would be good to add error bars or error estimation. There are aspects of the discussion that would benefit from this. This is particularly the case for the discussion on possible Li-losses – we know APT is not so good with oxygen:) – but for Li not much has been done. This would be a good resource for future studies. Please see detailed comments and references in the attached document. I think there might also be an issue with some of the references in the submitted manuscript.

Author reply to comment

We fully agree with the reviewer’s comment. We had added error bars, but they did not seem to be clearly visible. In our case, the error bars for Li range from about 0.2 to 0.4 at.%. Per the suggestion, we enlarged the error bars and error estimation for better visualization.

Unfortunately, it is difficult to explain the possible Li loss with our data. In our experience, APT analysis from coin cells fabricated at the laboratory scale showed a low rate of success, considerably variation, and rather large error bars. However, APT analyses from 18650 cells fabricated in pilot plants showed higher success, with relatively uniform results and smaller error bars, as explained above. In addition, a delithiated sample (fully charged) also showed lower success and larger error bars than the lithiated sample (fully discharged). Thus, with the current results, we only dealt with the Li distribution in the lithiated sample in this manuscript.

The following changes were made:

Please see Fig. 1c,

Please see Page #5

“We quantitatively analysed the compositions of NCM-pristine, NCM-formation, NCM-100 cycles, and NCM-300 cycles as a function of the radial direction from the surface of a secondary particle in contact with the liquid electrolyte, which acts as a pathway for the extraction and insertion of Li ions (Table 2, Fig. 1b–d, and Supplementary Fig. 4).”

Please see Supplementary Fig. 5b,

Specimen		Atomic ratio
		Li
NCM-pristine		1.03 ± 0.016
NCM-formation		0.96 ± 0.012
NCM-100 cycles	R1 (<100 nm)	0.57 ± 0.013
	R2 (~200 nm)	0.68 ± 0.024
	R3 (~300 nm)	0.76 ± 0.011
	R4 (~2 μm)	0.99 ± 0.035
NCM-300 cycles	R0 (surface)	0.03 ± 0.004
	R1 (<30 nm)	0.37 ± 0.011
	R2 (~100 nm)	0.47 ± 0.016
	R3 (~350 nm)	0.69 ± 0.013
	R4 (~700 nm)	0.77 ± 0.027
	R5 (~1 μm)	0.84 ± 0.042
	R6 (~1.5 μm)	0.91 ± 0.001
	R7 (~2 μm)	1.03 ± 0.018

Supplementary Fig. 5b. Atomic fraction of Li along the depth from the surface, as measured by APT.

Please see Fig. 2,

Please see Fig. 3a and d,

Please see Fig. 4a and d,

Please see Fig. 6a and d,

3. It is not completely clear how the authors estimated the depth at which the APT analysis was performed, or how accurate this was, yet this is important for the discussion.

Author reply to comment

We measured the depth when we fabricated the APT specimens by using a FIB lift-out method. APT samples were taken from the centre of the NCM cathode surface, as indicated by a yellow dotted rectangle in Supplementary Fig. 4. Then, the APT samples were fabricated as a function of the radial direction from the surface of the particle. The distance of the analysed location from the particle surface was measured in the FIB electron beam image. There may be some slight errors in the measured depth of the specimen fabricated for the centre of the particle, but we believe that the measured depths of the specimens fabricated near the surface are accurate. We added a cross-sectional SEM image and how we estimated the depth of the APT samples in Supplementary Fig. 4.

The following changes were made:

Please see Page #5,

“NCM particles are spherical with diameters of ~10 μm , and each NCM particle consists of hundreds of smaller (200–600 nm) primary particles (Supplementary Figs. 3 and 4).”

Please see Page #5,

“We quantitatively analysed the compositions of NCM-pristine, NCM-formation, NCM-100 cycles, and NCM-300 cycles as a function of the radial direction from the surface of a secondary particle in contact with the liquid electrolyte, which acts as a pathway for the extraction and insertion of Li ions (Table 2, Fig. 1b–d, and Supplementary Fig. 4).”

Please see Supplementary Fig. 4,

Supplementary Fig. 4. APT analysis regions in NCM. Image of a disassembled cell (a). Cross-sectional images of an NCM cathode fabricated by a cooling cross-section polisher (b) and an NCM particle sectioned by FIB (c). SEM image of the needle-shaped APT specimen after final milling (d). APT samples were taken from the centre of the NCM cathode surface, as indicated by the yellow dotted rectangle. Then, the APT samples were fabricated as a function of the radial direction from the surface of the particle, as indicated by the yellow arrows. The distance from the surface of each sample was measured by FIB.

4. I would recommend that the authors dig a bit deeper into their APT data. First at this stage they report on visual inspections of the data to assess the homogeneity of the distribution of Li and other species, which is not good enough IMHO unless it can be justified. Second, if the orientation of the sample is known, even approximately, it is likely that atomic planes can be imaged or searched for from within the data. Could be that a pole is visible? if the authors have trouble with this or making this clear they could seek support or help from other experts in the community (I would be happy to help search if necessary). This might provide additional information that could be valuable.

Author reply to comment

We are grateful for these helpful comments and agree with the reviewer's suggestion. We believe that the corresponding changes have made our manuscript more reliable.

First, we assessed the homogeneity of Li, Ni, and O (the main components) by using a frequency distribution analysis in IVAS software, following your advice. From the frequency distribution analysis, we reconfirmed that the homogeneity of the constituent elements was maintained in most cases that we discussed in the manuscript. The values of the Pearson coefficients (μ), a measure of the homogeneity of a distribution, indicate considerable homogeneity along the depth, as shown in Supplementary Fig. 7. The observed frequency distribution corresponds well to a binomial frequency distribution. In the case of the non-homogeneous distribution in Fig. 6, however, the value of μ for Li is higher than 0.9, and the observed frequency distribution significantly deviates from the binomial distribution (Supplementary Fig. 12). This finding indicates a statistically non-uniform distribution, which deviates from a random solid solution. The P values for Li, Ni and O were observed to be below 0.001. In addition, we show the main elements here, but the values of μ for Mn and Co were also lower than 0.05, meaning a high degree of uniformity. To clarify our APT data, we added the frequency distribution graph and the values of μ and P in Supplementary Figs. 7 and 12.

The following changes were made:

Please see Page #7,

“Further, as shown in Fig. 2b, Li (red), Ni (green), Co (violet), Mn (orange), and O (blue) were all very uniformly distributed in 3D without any deficiency or segregation. To estimate the homogeneity of the elemental distribution, we performed a frequency distribution analysis. The value of the Pearson coefficient (μ), an indicator of the homogeneity of a distribution, for each constituent element remained low. In addition, the frequency distributions were close to binomial distributions along the depth during cycling, as shown in Supplementary Fig. 7 for Li, Ni, and O, which reconfirms that the distribution is quite homogeneous.”

Please see Page #11,

“However, we also observed the non-uniform distribution of constituent elements at several sites after cycling, in agreement with previous reports on local compositional variations^{16,33,34}. The values of μ for Li are higher than 0.9, and the observed frequency distributions significantly deviate from binomial distributions, indicating a higher degree of non-uniformity, as shown in Supplementary Fig. 12.”

Please see Supplementary Fig. 7,

Supplementary Fig. 7. Compositional homogeneity according to a frequency distribution analysis. This analysis of NCM shows a rather uniform distribution of the constituent elements Li, Ni, and O along the depth from the particle surface. A Pearson coefficient (μ) far from 1.0 indicates a homogeneous distribution, and the observed frequency distribution is almost a binomial distribution.

Please see Supplementary Fig. 12,

Supplementary Fig. 12. Compositional homogeneity according to a frequency distribution analysis of inhomogeneous regions. Frequency distribution analysis of the inhomogeneous region in NCM shows non-uniform distributions of the constituent elements. A Pearson coefficient (μ) close to 1.0 means a high degree of non-uniformity.

Second, we agree with the reviewer's comment that the orientation might provide valuable information on the elemental distribution. A [100] pole is needed to observe the layered structure represented by (003) plane, but this pole was not visible in our data, as shown in Supplementary Fig. 15, which is common in the APT analysis of oxide materials. In contrast, Si clearly shows pole. We are also interested in this issue, and we plan to address it with a correlative STEM and APT approach in our next study.

The following changes were made:

Please see Page #14,

“A pole is not visible in our APT data, as shown in Supplementary Fig. 15.”

Please see Supplementary Fig. 15,

Supplementary Fig. 15. Detector event histogram of NCM and Si. Detector maps of NCM (a) and Si (b). A pole is not visible in the detector map of NCM, whereas a pole is clearly visible in that of Si

5. I think it is unnecessary to justify using APT in current times - especially not "against" STEM but to complement STEM. I commented on this in the attachment.

Author reply to comment

We fully agree with the reviewer's comment. Per your suggestion, we corrected the tone of the sentence in the manuscript that you commented on. Our intention is also to suggest a complementary approach of APT and STEM. It seems that the sentence was distorted, as we wrote it to explain APT to electrochemists who are unfamiliar with it. We are sure that APT is a useful and reliable technique for local quantitative analysis.

The following changes were made:

Please see Page #3,

“Among widely used experimental techniques, scanning transmission electron microscopy (STEM) and advanced X-ray techniques have provided invaluable information, such as the crystalline lattice structure^{17–20} and chemical states^{21–30}. Thus far, however, they lack the ability to directly quantify the charge carrier in LIBs, Li ions, at high spatial resolution.”

***Detailed comments on the written text**

1. This sentence serves no purpose.

Author reply to comment

We are grateful for the comment and agree with the reviewer's suggestion. We deleted the entire sentence.

The following changes were made:

Please see Page #3,

~~“Thus, no direct evidence supports the accepted degradation mechanism, and the behavior of Li ions remains to be clarified.”~~

2. I think the tone here should be changed – there is no reason to oppose APT and STEM – simply that the shortcomings of STEM can be compensated by the strength of APT and hence best combine them. The APT community has consistently tried to built its identity against TEM instead of a perfect complement to TEM and that has led to numerous conflicts that I think are unnecessary □.

Author reply to comment

Our reply to this comment is the same as our response to comment 5 above.

3. A bit redundant with the previous paragraph.

Author reply to comment

We intended to show the most widely used experimental techniques for battery research and to highlight their limitation. To accurately convey our intention to readers, we changed these sentences.

The following changes were made:

Please see Page #3,

“Among widely used experimental techniques, scanning transmission electron microscopy (STEM) and advanced X-ray techniques have provided invaluable information, such as the crystalline lattice structure^{17–20} and chemical states^{21–30}. Thus far, however, they lack the ability to directly quantify the charge carrier in LIBs, Li ions, at high spatial resolution.”

4. I think this was clear from the intro.

Author reply to comment

We are grateful for the helpful comment and agree with the reviewer’s suggestion. We deleted this entire sentence.

The following changes were made:

Please see Page #3,

~~“thus, mapping the Li ions at the sub nanometre scale in 3D is essential for a comprehensive understanding of the behaviour of the Li ions, especially related to changes in the atomic arrangements.”~~

5. MAYBE BEST STATE THIS IN THE INTRO (oops all caps were not intentional)

Author reply to comment

We agree with the reviewer’s suggestion. In addition, the sentence in this paragraph (results) was redundant with the sentence in the introduction. We deleted this sentence and changed the sentence in the introduction to use academically more correct terminology.

The following changes were made:

Please see Page #5,

~~“To achieve this aim, we employed STEM, which reveals the atomic arrangements, together with APT, which provides quantitative information on the constituent elements.”~~

Please see Page #3,

“To achieve this aim, we employed STEM, which reveals the atomic arrangements, together with atom probe tomography (APT) at the same location, which provides 3D quantitative information on the constituent elements, including Li ions, with a sensitivity of ~10 parts per million³¹⁻³⁴”

6. This has rarely been reported for Li

The mechanisms in Mancini’s paper were mostly all contradicted in

<https://iopscience.iop.org/article/10.1088/1367-2630/18/3/033031>

and Zanuttini explains why

<https://journals.aps.org/pr/abstract/10.1103/PhysRevA.95.061401>

but that’s for N/O that can easily form stable neutral atoms or molecules

for Li, this may not be the same issues – likely closer to issues related to multi-hit capability of the detector so maybe see:

<https://www.sciencedirect.com/science/article/pii/S0304399115300164?via%3Dihub>

or <https://www.sciencedirect.com/science/article/pii/S0304399117305272>

that discuss these issues in more details. 39 does not contain APT results...

Author reply to comment

We appreciate this very important point. Previously, we only thought of oxygen because Li (NCM)O₂ also includes it. We should have thought about the multi-hit events, as you advised. Per the suggestion, we revised the text and added the related references.

The following changes were made:

Please see Page #5,

“In the case of compound materials, however, the accuracy of APT’s compositional measurement depends on the analysis conditions such as the laser pulse energy³⁶⁻⁴¹.”

Please see Page #9,

“At low magnification, crack formation or an increased gap between primary particles was also observed after 300 cycles, as previously reported many times (Supplementary Fig. 10)⁴²⁻⁴⁵.”

Please see Page #10,

“While the Li ions move toward the anode during the charge process, TMs readily move to and occupy the 3b Li sites in the Li layer via the tetrahedral sites because the formation of a disordered structure is energetically favourable in the O3-type layered structures (Fig. 5a)^{14,46-48}.”

Please see Page #11,

“Although this study does not clarify the detailed relationship between the arrangement of atomic defects and Li accommodation sites, we could conclude that these defects change the local atomic arrangement and hence the lattice strain⁴⁹⁻⁵¹, and eventually the distribution of Li⁵¹.”

Please see **References**,

- [36] Devaraj, A., Colby, R., Hess, W. P., Perea, D. E. & Thevuthasan, S. Role of photoexcitation and field ionization in the measurement of accurate oxide stoichiometry by laser-assisted atom probe tomography. *J. Phys. Chem. Lett.* **4**, 993–998 (2013).
- [37] Silaeva, E. P., Karahka, M. & Kreuzer, H. J. Atom probe tomography and field evaporation of insulators and semiconductors: Theoretical issues. *Curr. Opin. Solid State. Mater. Sci.* **17**, 211–216 (2013).
- [38] Gault, B. et al. Behavior of molecules and molecular ions near a field emitter. *New J. Phys.* **18**, 033031 (2016).
- [39] Zanuttini, D. et al. Simulation of field-induced molecular dissociation in atom-probe tomography: Identification of a neutral emission channel. *Phys. Rev. A* **95**, 061401 (2017).
- [40] Meisenkothen, F. et al. Effects of detector dead-time on quantitative analyses involving boron and multi-hit detection events in atom probe tomography. *Ultramicroscopy* **159**, 101–111 (2015).
- [41] Peng, Z. et al. On the detection of multiple events in atom probe tomography. *Ultramicroscopy* **189**, 54–60 (2018).
- [42] Sun, H. H. & Manthiram, A. Impact of microcrack generation and surface degradation on a nickel-rich layered Li [Ni_{0.9}Co_{0.05}Mn_{0.05}]O₂ cathode for lithium-ion batteries. *Chem. Mater.* **29**, 8486–8493 (2017).
- [43] Park, K. J. et al. Degradation mechanism of Ni-enriched NCA cathode for lithium batteries: Are microcracks really critical? *ACS Energy Lett.* **4**, 1394–1400 (2019).
- [44] Kondrakov, A. O. et al. Anisotropic lattice strain and mechanical degradation of high- and low-nickel NCM cathode materials for Li-ion batteries. *J. Phys. Chem. C* **121**, 3286–3294 (2017).
- [45] Park, S. Y. et al. Probing electrical degradation of cathode materials for lithium-ion batteries with nanoscale resolution. *Nano Energy* **49**, 1–6 (2018).
- [46] Lim, J. M. et al. The origins and mechanism of phase transformation in bulk Li₂MnO₃: first-principles calculations and experimental studies. *J. Mater. Chem. A* **3**, 7066–7076 (2015).
- [47] Lee, E. & Persson, K. A. Structural and chemical evolution of the layered Li-excess Li_xMnO₃ as a function of Li content from first-principles calculations. *Adv. Energy Mater.* **4**, 1400498 (2014).
- [48] Reed, J. & Ceder, G. Role of electronic structure in the susceptibility of metastable transition-metal oxide structures to transformation. *Chem. Rev.* **104**, 4513–4534 (2004).
- [49] Li, Q. et al. Dynamic imaging of crystalline defects in lithium-manganese oxide electrodes during electrochemical activation to high voltage. *Nat. Commun.* **10**, 1–7 (2019).
- [50] Wang, R. et al. Atomic structure of Li₂MnO₃ after partial delithiation and re-lithiation. *Adv. Energy Mater.* **3**, 1358–1367 (2013).
- [51] Singer, A. et al. Nucleation of dislocations and their dynamics in layered oxide cathode materials during battery charging. *Nat. Energy* **3**, 641–647 (2018).

7. vague – what do you mean? Based on what criteria? Maybe report in SI how you did this in

Author reply to comment

Early in this work, we referred to previous atom probe studies on a cathode material. A. Devaraj quantified a $\text{Li}[\text{Li}_{0.2}\text{Ni}_{0.2}\text{Mn}_{0.6}]\text{O}_2$ cathode material with a laser pulse energy of 20 pJ¹, and J. Y. Lee quantified $\text{Li}(\text{Ni}_{1/3}\text{Co}_{1/3}\text{Mn}_{1/3})\text{O}_2$ cathode material with a laser pulse energy of 30 pJ². In particular, J. Y. Lee showed that the composition of the cathode material did not significantly vary under 30 pJ. Based on these previous studies and our ICP-AES results on NCM-pristine, a laser pulse energy of 25 pJ and a base temperature of 30 K were selected as our experimental conditions. We also conducted APT experiments under various conditions and confirmed the tendency, which was observed in the previous studies. Unfortunately, because our company's policy is to delete old data, the data for the optimization in the early date of the study disappeared and could not be attached here.

[1] Devaraj, A. et al. Visualizing nanoscale 3D compositional fluctuation of lithium in advanced lithium-ion battery cathodes. *Nat. Commun.* **6**, 1-8. (2015).

[2] Lee, J. Y. et al. Three-dimensional evaluation of compositional and structural changes in cycled $\text{LiNi}_{1/3}\text{Co}_{1/3}\text{Mn}_{1/3}\text{O}_2$ by atom probe tomography. *J. Power Sources* **379**, 160-166. (2018).

8. This is a better terminology than normalised concentration

Author reply to comment

We agree with the reviewer's suggestion. We changed the "normalised atomic concentration" to "Atomic ratio".

The following changes were made:

Please see Table 2,

“Table 2. Site-specific quantitative APT results along the depth of pristine Li(Ni_{0.80}Co_{0.15}Mn_{0.05})O₂ (NCM) and NCM subjected to formation, 100 cycles, and 300 cycles (the atomic ratio is normalised to obtain a total metal fraction of 1).”

Specimen		Atomic ratio				
Process	Region (depth from the surface)	Li	Ni	Co	Mn	O
NCM-pristine	R (from surface to centre)	1.03	0.84	0.12	0.04	1.94
NCM-formation	R (from surface to centre)	0.96	0.81	0.15	0.05	1.89
NCM-100 cycles	R1 (<100 nm)	0.57	0.79	0.16	0.05	1.85
	R2 (~200 nm)	0.68	0.79	0.16	0.05	1.90
	R3 (~300 nm)	0.76	0.79	0.16	0.05	1.73
	R4 (~2 µm)	0.99	0.79	0.16	0.05	1.92
NCM-300 cycles	R0 (surface)	0.03	0.81	0.16	0.03	1.63
	R1 (<30 nm)	0.37	0.81	0.14	0.05	1.88
	R2 (~100 nm)	0.47	0.83	0.13	0.04	1.84
	R3 (~350 nm)	0.69	0.81	0.14	0.05	1.86
	R4 (~700 nm)	0.77	0.82	0.14	0.04	1.79
	R5 (~1.0 µm)	0.84	0.80	0.15	0.05	1.91
	R6 (~1.5 µm)	0.91	0.82	0.14	0.04	1.94
	R7 (~2.5 µm)	1.03	0.82	0.14	0.04	1.91

9. I think what appears to be missing here is the error estimation from the counting of atoms in the APT. I think this should maybe be mentioned. Was the isotopic ratio of Li been checked to assess possible pile-up effects?

Author reply to comment

We had already added error bars, not in Table 2, but in Fig. 1c. Per this suggestion, we added the error estimation for Li to Supplementary Fig. 5b because it was too complicated to add to Table 2. The ratio of ${}^6\text{Li}^{1+}$ and ${}^7\text{Li}^{1+}$ is almost the same as the ideal isotopic ratio, as shown in Supplementary Fig. 5a. Therefore, the pile-up effect does not seem to be large as only a negligible amount of Li deviates from the isotopic ratio.

The following changes were made:

10. A concentration is typically defined (IUPAC) as a quantity per unit volume, here you're reporting a dimensionless atomic fraction, might be best to be consistent with wording.

Author reply to comment

We thank the reviewer for this helpful comment, which we fully agree with. For academically accurate terminology, we corrected “Li concentration” to “amount of Li” or “atomic fraction of Li” when we directly referred to the specific atomic fraction in our APT data. (When we refer the overall trend of Li, we used “concentration”.) In addition, we changed “atomic concentration” to “atomic ratio” in the manuscript and figures.

The following changes were made:

Please see Page #5,

“However, the cycled NCM showed a substantial Li deficiency, **with an atomic fraction of Li less than 1.00**, except for the most central regions of the particle.”

Please see Page #6,

“Specifically, **the atomic fraction of Li** decreased significantly from 1.03 for NCM-pristine and 0.96 for NCM-formation to 0.57 for R1 in NCM-100 cycles.”

Please see Page #6,

“The results of the quantitative APT analysis thus prove that a significant Li loss occurs at the top surface and that **the amount of Li** increases with increasing depth from the top surface of the particle.”

Please see Pages #6–#7,

“**The atomic fraction of Li** at the top surface (R0) and at depths of ~30 nm (R1), ~100 nm (R2), ~350 nm (R3), ~700 nm (R4), ~1 μm (R5), ~1.5 μm (R6), and ~2.5 μm (R7) below the top surface are 0.03, 0.37, 0.47, 0.69, 0.77, 0.84, 0.91, 1.03, respectively. Thus, as clearly shown in Fig. 1c and Table 2, **the amount of Li** gradually increased from the surface to the centre of the particles in both NCM-300 cycles and NCM-100 cycles. However, in NCM-300 cycles, **the Li-depleted region extended** beyond ~1.5 μm below the surface. In addition, at a similar depth, **the amount of Li** in NCM-300 cycles is noticeably lower than that in NCM-100 cycles, revealing that the Li concentration gradient extends deeper in NCM-300 cycles owing to an increase in the Li-depleted area.”

Please see Page #7,

“The APT measurements near the surface of NCM-pristine provided **atomic ratios** that were in good agreement with the data determined by ICP-AES (Fig. 2a).”

Please see Page #8,

“By contrast, the STEM and APT results of NCM-300 cycles reveal distinct changes in the atomic structures and the corresponding amount of Li as a function of the depth. At the top surface (R0) of NCM-300 cycles, the atomic ratio determined by APT is Li:Ni:Co:Mn:O = 0.03:0.81:0.16:0.03:1.63 (Fig. 3a and Table 2), exhibiting the lowest amount of Li. Below the upper ~15 nm region, the amount of Li gradually increased with the increasing distance from the surface, which implies that this region consists mainly of the TMs and oxygen, along with the most significant Li deficiency (Fig. 3a).”

Please see Pages #8–#9,

“At ~30 nm from the top surface (R1), the atomic fraction of Li is higher than that at the top surface (R0), which shows the value of 0.37 (Fig. 3d). The depletion of Li in R1 was suppressed by the decrease in the migration of TMs into 3b Li sites. Clearly, the TMs migrated to a lesser degree than they did near the top surface, as denoted by red arrows (Fig. 3e and Fig. 3f). These stark differences in both the Li concentration and the structural disordering at the surface of NCM-300 cycles was mitigated in the interior of the particle. Nevertheless, at ~700 nm from the top surface (R4), the APT depth profile still shows a slightly deficient amount of Li (Fig. 4), but this deficiency is markedly suppressed relative to that near the surface (R0–R3), showing an atomic fraction of Li greater than ~0.70.”

Please see Page #9,

“Indeed, the central region of the particle presents almost the same amount of Li as that of NCM-pristine (Fig. 4d).”

11. Unclear what this refers to??

Author reply to comment

We thank the reviewer for the helpful comment. We intended to explain that the Li concentration gradient from the surface to the centre is formed in both NCM-100 cycles and NCM-300 cycles. To avoid the confusion, we change the sentences.

The following changes were made:

Please see Page #7,

“Thus, as clearly shown in Fig. 1c and Table 2, the amount of Li gradually increased from the surface to the centre of the particles in both NCM-300 cycles and NCM-100 cycles. ~~Moreover, the consistency of the Li distribution between NCM-100 cycles and NCM-300 cycles confirms a Li concentration gradient formed in the radial direction of the particle after cycling.~~”

12. At this stage, it is not clear to me how this depth was determined or what is the error on this estimation.

Author reply to comment

Our reply to this comment is the same as our response to comment 3 above.

13. Well... yes but this could have been an average loss, while the APT is very local, it's unclear to me that these results can be directly compared actually. Trends likely yes but might be best to state this clearly.

Author reply to comment

We thank the reviewer for the comment. We also want to compare the trends in the Li loss, because the ICP-AES data shows only an average loss of Li during cycling. Per this suggestion, we additionally stated this finding for clarity.

The following changes were made:

Please see Page #7,

“The APT measurements near the surface of NCM-pristine provided concentration values that were in good agreement with the data determined by ICP-AES (Fig. 2a). The trends in the Li loss during cycling in the APT and ICP-AES results are also similar, showing a greater Li loss after 300 cycles”.

14. Please perform statistical analyses of the Li distribution from the APT data. Unclear what the spatial resolution of APT would be in such a case, but at this stage, it's purely visual inspection and I think it makes the paper weaker.

Author reply to comment

Our reply to this comment is the same as our response to comment 5 above.

15. This is not performed on the atom probe specimen, is it? this should be made crystal clear.

Author reply to comment

The STEM-HAADF imaging of the layered structure is not performed on the atom probe specimen, as you deduced. We revised the sentence to correctly convey our methods.

The following changes were made:

Please see Pages #7–#8,

“The STEM–high angle annular dark field (HAADF) image of NCM-pristine from the identical location also shows a clear layered structure (R-3m) represented by alternating bright TM interlayers and dark Li interlayers, both at the surface of and inside particles (Fig. 2c and 2d, respectively).”

16. What would be the sensitivity limit of this measurement? it may be that this happened but cannot be detected/seen. I think this needs to be estimated and reported very carefully.

Author reply to comment

We are grateful for the helpful comment and fully agree with the reviewer's advice. A small amount of the TMs that migrate into Li layer cannot be seen in the HAADF intensity profile, as you pointed out. Our expression was unclear and confusing in the sentence, so we revised it accordingly. Unfortunately, it is difficult to specify the sensitivity limit of the HAADF intensity profile because the intensity of the peak would be affected by the thickness of TEM sample. The trend in TM migration can be seen, but it is difficult to quantify.

The following changes were made:

Please see Page #8,

“In the intensity profile along the (003) plane, the discrete regions of strong intensity represent the TM layer with a d-spacing of 0.47 nm (Fig. 2e), indicating that TMs rarely migrate into the Li layer.”

Please see Page #9,

“Even after 300 cycles, the layered structure is well maintained in the central region of the particle, with almost no TM migration (Fig. 4e and 4f).”

17. What is the orientation along which the APT data was acquired? If similar then the plane spacing seems wide, and maybe atomic planes can also be visualised in the APT data if the analysis is performed carefully. I think this could potentially provide some valuable complementary data.

Author reply to comment

Our response to this comment is the same as our reply to comment 4 above.

18. Can this be quantified?

Author reply to comment

It is difficult to quantify the degree of TM migration and capacity for Li accommodation. Because the HAADF intensity profile is affected by the thickness of the TEM sample, we performed a relative comparison between the locations in this work. STEM-annular bright field imaging has been applied to cathode materials to detect Li ions, but quantifying these Li ions and the disorder is difficult. Quantitative STEM (QSTEM) simulation, which is used for quantitative STEM imaging, is a useful technique for analysing 2D materials such as MoS₂. Even with this method, however, the quantification of the bulk specimen is difficult.

19. That graph is very hard to read...

Author reply to comment

As you stated, the graph in the schematic illustration can be misunderstood because we overlapped the 2D depth profile with the 3D schematic. In addition, the extent of Li depletion is already expressed by the red contrast. Thus, we removed the overlapping depth profile.

The following changes were made:

20. Could be associated to local atomic neighbourhood too, not just strain? This is something that could also maybe come from DFT-type calculations??

Author reply to comment

The shape of the Li distribution observed by APT is similar to that of the line defects often observed by STEM. In addition, a change in the displacement field has been recently reported, as we mentioned in our manuscript. This is why we suspect that the inhomogeneous Li distribution is caused by defects. As you mentioned in the comment, the local atomic neighbourhood, not just the strain, can affect the Li

distribution. However, it seems that there are too many variables to consider when analysing the Li distribution with such a complex structure and composition in DFT-type calculations. We believe that these considerations might be dealt with separately from the current paper, and in fact, we are planning to do this as one of our next research directions.

Reviewer #3 (Remarks to the Author):

This manuscript provides scanning transmission electron microscopy (STEM) and atom probe tomography (APT) data on $\text{Li}(\text{Ni}_{0.80}\text{Co}_{0.15}\text{Mn}_{0.05})\text{O}_2$ (NCM) oxides. The samples are from a pristine electrode and from oxide electrodes from cells that underwent 100 and 300 cycles in NMC/Gr cylindrical 18650 cells. The authors correlate changes in the local Li composition in the oxide (obtained by APT) and the structural changes (obtained by STEM) to cell capacity degradation on cycling. The paper is written well and relatively easy to read. The use of STEM to examine structural changes on cycling is not novel and has been used for over a decade; reference 10 in the manuscript gives one such example. The finding of rock-salt structure on the oxide surface is also not new; such data were reported by Abraham et al. [Electrochemistry Communications 4 (2002) 620–625] about 18 years ago.

Author reply to comment

As the reviewer commented, the rock-salt-like structure is now well known and has been proven to be a mechanism responsible for cathode degradation. However, this rock-salt-like structure is not a principal topic of this manuscript. We focused on the direct correlation between structural variations and the Li composition, which has not been experimentally proven yet.

The use of APT to study oxide degradation on cycling can be considered novel. APT has been used previously to study other battery oxides, e.g. [<https://doi.org/10.1038/ncomms9014>]. Although the technique is interesting, interpretation of the results is difficult because of artefacts that could be introduced during the sample preparation/measurement process. Here, the authors appear to have done their best to “calibrate” the data, based on results obtained on the pristine sample. However, in the opinion of this reviewer, the data appear to be inconsistent and, in some ways, incomplete.

Author reply to comment

Yes, the originality of our contribution lies in the site-specific Li quantification to prove a direct relationship between the atomic structure and Li accommodation capability. As the reviewer mentioned, APT is a very delicate technique. We truly understand the reviewer’s concerns. After many debates in the APT, physics, and chemistry communities, it has now been verified that APT can be fully utilised as a complementary technique to TEM/STEM analyses. To ensure the consistency of our APT experiments, all fabrication processes including electrode synthesis, assembly, cycling, and disassembly were conducted under strict protocols in our pilot plants. Based on our experience, APT analysis from

18650 cells is quite reliable, but APT from coin cells (half type) fabricated at the lab scale shows considerable variation and is consequently not reliable, as in the publication that the reviewer provided as an example.

Some questions and concerns about the manuscript are as follows:

About the Experimental –

1. How many formation “cycles” were conducted? Only 1 cycle is shown in Figure S2.

Author reply to comment

Two cycles were conducted for formation, as mentioned in the Electrochemical Measurements section in the Methods. Previously, data from the first formation step were shown in Supplementary Fig. 2, which illustrated the initial resistance of the cells. Electrochemistry data for two cycles of formation are shown below. We changed Supplementary Fig. 2 to show the electrochemistry data of the second formation step as it represents more general charge–discharge behaviour.

Electrochemistry data from two formation cycles.

The following changes were made:

Please see Supplementary Fig. 2,

Supplementary Fig. 2. Electrochemical cycling performance of NCM. Charge and discharge curves of an NCM cell operated in the voltage range of 2.8–4.35 V at 0.2C and 45 °C.

2. What is the thickness of the oxide electrodes used in the study? From which part of the electrode were the particles selected for study? From the electrode surface? Oxide particles closest to the electrode surface are expected to have a higher degree of degradation than those closer to the current collector. See Gilbert et al. [Adv. Mater. Interfaces 2018, 5, 1701447] and Yang et al. [Adv. Energy Mater., vol. 9, no. 25, p. 1900674, Jul. 2019.]

Author reply to comment

We appreciate your important point-out. The oxide electrodes are ~55 μm thick. The sampling position in the electrode must be consistent among the cells. Therefore, our STEM and APT specimens were always fabricated from a secondary particle close to the surface. The sampling positions are described in detail in Supplementary Fig. 4.

The following changes were made:

Please see Supplementary Fig. 4,

Supplementary Fig. 4. APT analysis regions in NCM. Image of a disassembled cell (a). Cross-sectional images of an NCM cathode fabricated by a cooling cross-section polisher (b) and an NCM particle sectioned by FIB (c). SEM image of the needle-shaped APT specimen after final milling (d). APT specimens were taken from the centre of the NCM cathode surface, as indicated by the yellow dotted rectangle. Then, the APT samples were fabricated as a function of the radial direction from the surface of the particle, as indicated by the yellow arrows. The distance from the surface of each sample was measured by FIB.

3. Were the oxide electrodes “harvested” with the cell in the discharged state? Were these electrodes washed before APT and STEM sample preparation?

Author reply to comment

Yes, the cells were disassembled in the discharged state. According to the strict safety protocols of our company, especially to avoid the risk of ignition, large cells such as 18650 -type cells should be disassembled only in the discharged state. There was no additional washing. For clarity, the following phrase has been added to the Electrochemical Measurements section in the Methods.

The following changes were made:

Please see page #13,

“All cells were disassembled in a humidity-controlled room after being discharged.”

4. Were the APT samples prepared in air? If so, could there be Li loss from the surface regions because of reaction with the ambient? These reactions would be pronounced for the delithiated oxides that are extracted from the cycled electrodes.

Author reply to comment

APT samples were prepared by using FIB, which operates in a high vacuum (10^{-5} Pa). Immediately after FIB sample preparation, the samples were loaded into the APT instrument, which also operates in an ultra-high vacuum (10^{-11} Pa). Thus, we believe that there was no Li loss during sample preparation using FIB and APT analysis. Additionally, APT data acquired from the outermost sample surface were not counted to avoid possible surface changes, such as contamination and oxidation. These processes are routinely conducted in a general APT analysis, so they are not described in our manuscript.

5. What is the probe size for the APT experiments? Is the data from one primary particle or several primary particles? The authors indicate that these primary particles are 200-600 nm in size.

Author reply to comment

The APT sample probe has a diameter less than 100 nm and a length of ~400 nm, as shown in Supplementary Fig. 4. When the APT sample contains several primary particles, APT analysis could not be successfully done owing to the field concentration at the boundaries. This failure is generally pronounced in samples prepared by powder sintering such as LIB cathodes. Therefore, all APT data

were acquired from single primary particles. To ensure the accuracy of the data, we used the average value from five successful APT experiments at each position. The error bars of the APT data in Fig. 1c were enlarged for better visibility.

The following changes were made:

Please see Fig. 1c,

Please see Supplementary Fig. 4,

About the Results –

1. The authors appear to correlate capacity fade of the cell with changes in the local Li composition of the oxide, which is incorrect. Capacity fade of a NMC/Gr cell results from loss of mobile Li-ions to the solid electrolyte interphase (SEI) of the graphite electrode. These “lost” ions are from the oxide and hence the oxide becomes more Li-deficient on cycling, as the authors have shown. Loss of mobile Li-ions is the “definition” of capacity fade. However, this loss is due to electrolyte reduction reactions at the graphite electrode.

Author reply to comment

As the reviewer suggested, the largest cause of the capacity fade in NMC/Gr is known to be SEI formation. In our general tests (thousands of test cells over a period of five years), this is true. However, cathode degradation is still responsible for a substantial proportion of the capacity decay of the entire battery. According to our internal protocols, we randomly make coin cells from disassembled cathodes after long-term cycling and a fresh Li anode to check for a net capacity loss in the cathode. These tests have confirmed that 20–30% of capacity loss is due to cathode degradation, depending on fabrication/operation conditions. In addition, many reports have shown that simply modifying/improving the cathode enhances the cycling performance of full-type NCM/Gr cells. Otherwise, research on the cathode would not be necessary.

2. The NCM particles studied here are spherical with diameters of ~10 μm . The APT data are obtained up to a depth of 2.5 microns from the secondary particle surface. In both the 100-cycle and 300-cycle samples, below a depth of 2.5 microns, the oxide concentration is similar to that of the pristine oxide (Lines 128-129; 136-137). Because the oxide particles would have a 5 micron radii, these APT data would suggest that the particles between 2.5 and 5 microns are essentially inactive. Why? Because, once extracted, it is hard to put the Li-ions back into the oxide particles – this phenomena described as an “apparent” first cycle irreversibility has been described by Dahn et al [J Electrochem. Soc 147:3598 (2000) & Kang et al. [Mater Sci (2008) 43:4701–4706]. That is, the APT data suggest inactive particles in the oxide, which is incorrect because all particle appear to participate in the electrochemical reaction (as suggested by Figure S2).

Author reply to comment

Layered cathodes are well known to suffer from an enormous capacity loss during the first cycle. As the reviewer mentioned, the irreversibility has been widely studied. The formation of inactive domains, the local collapse of the interslab space, and the formation of solid-electrolyte interphases have been suggested as the origins of this irreversibility. In particular, Dahn and Kang investigated overlithiation in the surface region below 2 V and revealed that irreversibility can be suppressed with subsequent

relaxation. However, we assumed that the irreversible lithium vacancies in our cathodes were distributed inside the entire particle as we cycled the samples in a normal voltage window, and lithium ions can be distributed into the powder by grain boundary diffusion. It should also be mentioned that the lithiation state was homogeneous for a discharged single-crystal particle (NCM) after charge/discharge¹. We believe that the slight change in the lithium concentration owing to irreversibility during the first cycle was not sufficient to be detected in the APT analysis; thus, it is unlikely to contribute to the significant concentration difference between the pristine and long-term cycled cells.

In order to explore the dynamic evolution during the first cycle, new experiments using a cryo-setup to prevent additional Li migration after intended depth of charging/discharging must be designed. This reviewer's comment will be a good theme for the next investigation using our test methods.

[1] Zhang, F. et al. Surface regulation enables high stability of single-crystal lithium-ion cathodes at high voltage. *Nat. Commun.*, **11**, 3050. (2020).

3. **Lines 135-138** state the following: "The Li concentration at the top surface (R0) and at depths of ~30 nm (R1), ~100 nm (R2), ~350 nm (R3), ~700 nm (R4), ~1 μ m (R5), ~1.5 μ m (R6), and ~2.5 μ m (R7) below the top surface are 0.03, 0.37, 0.47, 0.69, 0.77, 0.84, 0.91, 1.03, respectively. Thus, as clearly shown in Fig. 1c and Table 2, the Li concentration gradually increased from the surface to the center of the particles." These data indicate that the Li sites in oxide particles close to the interior are full while those close to the exterior are empty. That is, there are high-impedance particles at the secondary particle surface and low-impedance particles close to the interior. If this is the case, how do the Li-ions reach the secondary particle interior? Are all secondary particle "cracked", so that the electrolyte can penetrate into the interior?

Author reply to comment

This is another very good question, which we also asked. Several reasons why Li ions still diffuse, despite the high impedance of the particles, can be expected. First, as pointed out by the reviewer, the electrolyte may permeate the cathode through the micro-cracks and grain boundaries as the path for Li ion penetration. Second, grain boundaries between primary particles are high-diffusivity paths for Li ions. This grain boundary effect has been reported in many publications such as [S. Han et al. / *Journal of Power Sources* 240 (2013) 155–167]. Another reason may be the slow C-rate, which allows enough time for Li ions to diffuse through the high-impedance surface region. A slower C-rate is well known

to result in a higher cell capacity, even in the same sample. Unfortunately, our study cannot explain all of these questions related to the charge/discharge dynamics in LIBs.

4. The Li concentration in Table 2 has been normalized so that the total metal fraction is 1. The proportion of Ni, Co and Mn are kept the same. How does the APT data account for the transition metal dissolution that happens from the oxide?

Author reply to comment

We re-measured the ICP-AES data with freshly disassembled 18650 cells in order to confirm the TM migration issue. The measured concentrations both in weight percent and the normalised atomic ratio are displayed in Table 1. After double-checking, we confirmed that there was no notable TM dissolution in these experiments. Our APT data also shows that TM dissolution was negligible, which is consistent with the ICP-AES data.

Although TM dissolution is a general phenomenon that degrades cathode performance, the amount of dissolved TMs varies depending on the cell fabrication conditions, voltage range, C-rates, depth of charge/discharge, and so on. We adopted very stable conditions and verified voltage windows, which are standardised for commercialization. The target of this manuscript was not to analyse LIB cells under harsh conditions such as fast charging or high-temperature operation.

The following changes were made:

Please see Table. 1,

Specimen	Atomic concentration (weight %)			
	Li	Ni	Co	Mn
Pristine	1.04 (6.77)	0.80 (44.31)	0.15 (8.24)	0.05 (2.46)
100 cycled	0.94 (6.16)	0.80 (44.36)	0.15 (8.23)	0.05 (2.46)
300 cycled	0.83 (5.44)	0.80 (44.58)	0.15 (8.29)	0.05 (2.48)

Table 1. Compositions of cationic elements in pristine $\text{Li}(\text{Ni}_{0.80}\text{Co}_{0.15}\text{Mn}_{0.05})\text{O}_2$ (NCM) and NCM subjected to 100 and 300 cycles as measured by ICP-AES (the atomic concentration is normalised to obtain a total TM fraction of 1).

Please see page #4,

“In addition, inductively coupled plasma-atomic emission spectroscopy (ICP-AES) confirms a significant loss of Li from the cathode after battery cycling (Table 1).”

Please see page #4,

“However, the Li concentration decreased dramatically with the number of cycles, whereas the composition of the TMs remained almost the same, as shown in Table 1.”

5. About the STEM results – The scale bar in the STEM figures is 2 nm; the data shown is from an area that is about 10 nm x 10 nm. Note here that the primary particle size is 200 – 600 nm. If all primary particles at the surface of the 10 micron size secondary particle have very little Li, then Figure 3 (300 cycle oxide) would show that every portion of the oxide particle has the rock-salt structure: not just the 10 top nm, but the entire 200 nm would be rock-salt.

Author reply to comment

First of all, our STEM results are very consistent with many other reported STEM results [see references 9–12, 13–15]. Although primary particles near the surface of secondary particle surface after 300 cycles exhibit severe cation mixing, 37–47% of Li ions remain in the lattice (positions R1 and R2 in Table 2). This means that the entire primary particle is not in the rock-salt-like structure. Only the top 10 nm near the surface shows a fully cation-mixed structure in the STEM image and less than 5% Li contents (Fig.

3a–c). In addition, the definition of rock-salt structure in layered oxide cathodes is not clear yet. The original rock salt structure should be a cubic phase, but no cation-mixed structures observed thus far have been completely cubic. That is why we and other authors have called this a “rock-salt-like” structure. We believe that the degree of phase transition must be strictly changed into the degree of cation mixing. However, we do not intend to argue this generally accepted definition. Gradual changes in the degree of cation mixing within primary particles has also been described in recent publications¹.

[1] Kang, Y. S. et al. Revealing the structural degradation mechanism of the Ni-rich cathode surface: How thick is the surface? *J. Power Source*. **490**, 229542. (2021).

6. The point here is that the STEM data and the APT data are on different scales: the STEM results are on a nanometer scale, whereas the APT data are on a micron scale. It may not be appropriate to correlate (& draw broad conclusions from) the data because they provide different types of information, which are complementary. This is the problem with the schematic in Figure 5. The atomic-scale STEM data is used to justify the micron-scale APT data.

Author reply to comment

The field of view of STEM micrographs ranges from 30×30 nm to 100×100 nm, depending on the magnification with a resolution of $1\text{k} \times 1\text{k}$ to $4\text{k} \times 4\text{k}$. For better visibility, the magnified areas were cropped. Examples of the original STEM images before cropping were added as Supplementary Fig. 13. In addition, the APT data are never on the micrometre scale. The APT specimens were conical, with a diameter of 20–40 nm and a length of ~ 400 nm. The regions of interest (ROIs) in both STEM and APT are very close to each other. Therefore, STEM and APT have been recognised as complementary techniques without any objections.

The following changes were made:

Please see page #13,

“To better visualise the structure, the region of interest is magnified and cropped (Supplementary Fig. 13).”

Please see Supplementary Fig. 13,

Supplementary Fig. 13. Examples of original STEM images before cropping. At low magnification, the structure is not clearly visible.

About the Discussion –

1. This paragraph is simply a restatement of the results and not an explanation/discussion of the data.

Author reply to comment

We appreciate this very important point. This was our mistake. The “Discussion” section now starts from line 204 and the “Conclusion” section summarizing our paper starts from line 257. The headings have been changed in the manuscript.

The following changes were made:

Please see Page #9,

“Discussion

Li accommodation in the layered structure”

Please see Page #12,

“Conclusions”

In summary, the STEM data shown are reasonable and consistent with similar information shown in the research literature. The electrochemistry data and the SEM images are also ok. However, the APT data may contain artefacts that distort the data. It is likely that the results will not be reproducible. Moreover, studies on a few oxide particles are not representative of a cycled electrode because of the inhomogeneity that results from the lithium extraction/insertion cycling. Hence, the conclusions, as depicted in the schematic of Figure 5, are likely incorrect.

Author reply to comment

Overall, we understand the reviewer’s concerns that APT data may not be reproducible and hence cannot represent general phenomena. However, APT analysis is becoming more popular in many areas of science, and it has been accepted as a very useful approach for acquiring quantitative elemental information in a nanoscale area. In terms of reproducibility, sample homogeneity is quite important. In order to overcome this homogeneity issue, we used mass-production facilities and 18650 full cells. In addition, several APT analyses were conducted at consistent locations for each sample. These approaches enabled us to acquire reliable APT results. In fact, we also tried an APT analysis on half-coin cells fabricated at the laboratory scale. The APT results from these cells showed a non-uniform distribution of Li and even other TMs, as in the paper suggested by the reviewer as an example. In order to enhance the reliability of APT data, we added more information to the Supplementary Information, which is offered as checkpoints for verifying the success of our analysis to the APT-related community.

In this research, we measured compositional variations in layered oxide cathodes during the long-term cycling of LIBs. Many questions remain about the acting mechanism within a single charge/discharge

step and phenomena that occur during standby. We believe that our local compositional analysis through APT to include Li ions will provide useful information for further cathode development and will suggest areas for further study in the research community.

Reviewer #1 (Remarks to the Author):

All of my comments were addressed well. This is some very good work.

Reviewer #2 (Remarks to the Author):

I'm satisfied with the changes done to the manuscript in response to my comments

Reviewer #3 (Remarks to the Author):

Satisfactory responses have been provided to reviewer questions and appropriate changes have been made to the manuscript.

The manuscript may be published in its current form.